# Shape of Thought: Progressive Object Assembly via Visual Chain-of-Thought

Yu Huo [* 1]  Siyu Zhang [* 1]  Kun Zeng [* 2]  Haoyue Liu [1]  Owen Lee [3]  Junlin Chen [1]
Yuquan Lu [2]  Yifu Guo [2]  Yaodong Liang [4]  Xiaoying Tang [† 1 5 6]

## Abstract

Multimodal models for text-to-image generation have achieved strong visual fidelity, yet they remain brittle under compositional structural constraints—notably generative numeracy, attribute binding, and part-level relations. To address these challenges, we propose **Shape-of-Thought (SoT)**, a visual CoT framework for *process-supervised progressive shape assembly in the rendered 2D domain*, without external engines at inference time. SoT trains a unified multimodal autoregressive model to generate interleaved textual plans and rendered intermediate states, helping the model capture shape-assembly logic without producing explicit geometric representations. Unlike text-only CoT, each decision is grounded in a rendered state, making counts, attachments, topology, and intermediate part-addition errors inspectable across the trajectory. To support this paradigm, we introduce **SoT-26K**, a large-scale dataset of grounded assembly traces derived from part-based CAD hierarchies, and **T2S-CompBench**, a benchmark for evaluating structural integrity and trace faithfulness. Fine-tuning on SoT-26K achieves 88.4% on component numeracy and 84.8% on structural topology, outperforming direct generation by +24.2 points on component numeracy and +19.3 points on structural topology. SoT establishes a transparent testbed for rendered-domain structure-aware generation. The code is available at https://github.com/yuhuo03/Shape-of-Thought.

---

[*]Equal contribution [†]Corresponding author [1]School of Science and Engineering, The Chinese University of Hong Kong, Shenzhen [2]Sun Yat-sen University [3]School of Data Science, The Chinese University of Hong Kong, Shenzhen [4]The Hong Kong University of Science and Technology, Guangzhou [5]Shenzhen Future Network of Intelligence Institute (FNii-Shenzhen) [6]Guangdong Provincial Key Laboratory of Future Networks of Intelligence, CUHK(SZ). Correspondence to: Xiaoying Tang <tangxiaoying@cuhk.edu.cn>.

*Proceedings of the 43rd International Conference on Machine Learning*, Seoul, South Korea. PMLR 306, 2026. Copyright 2026 by the author(s).

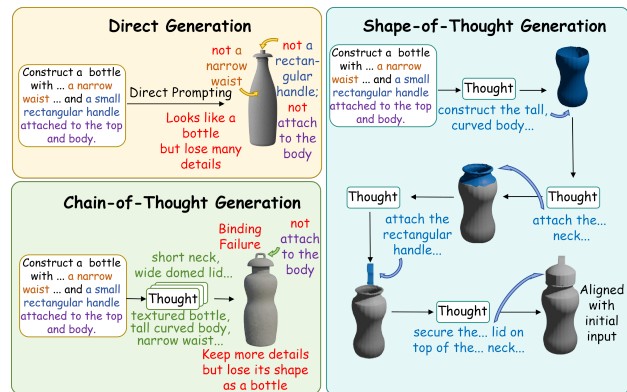

*Figure 1.* **Comparison of generation paradigms.** While direct generation fails to capture details and text-based CoT leads to semantic binding failures, SoT improves structural compliance on our tasks. By decomposing complex prompts into sequential visual sub-goals, SoT corrects structural deficiencies and ensures the final output aligns with the text description.

## 1. Introduction

Chain-of-Thought (CoT) prompting has fundamentally enhanced complex reasoning by eliciting intermediate rationales instead of relying on direct outputs (Wei et al., 2022; Wang et al., 2023). In multimodal settings, augmenting these traces with "visual thoughts" provides a powerful mechanism for spatial planning, effectively serving as a **visual working memory** (Yao et al., 2023; Li et al., 2025).

However, 2D generators remain brittle under compositional structural constraints, notably generative numeracy, attribute binding, and part-level relations. They often hallucinate counts or omit repeated thin structures when prompts demand fine-grained structural depiction (Ho et al., 2020; Deng et al., 2025). Meanwhile, text-only CoT can expand reasoning traces but still lacks visual grounding, making it hard to verify and correct structural drift along the generation trajectory. These failure modes are also relevant to view-based text-to-3D pipelines, where many modern systems rely on image-space priors and rendered-view supervision. As a result, compositional ambiguities and errors in the 2D guidance signal can propagate into 3D outputs as missing parts, wrong numeracy, or inconsistent relations (Hong et al., 2024). As illustrated in Figure 1, our method addresses these

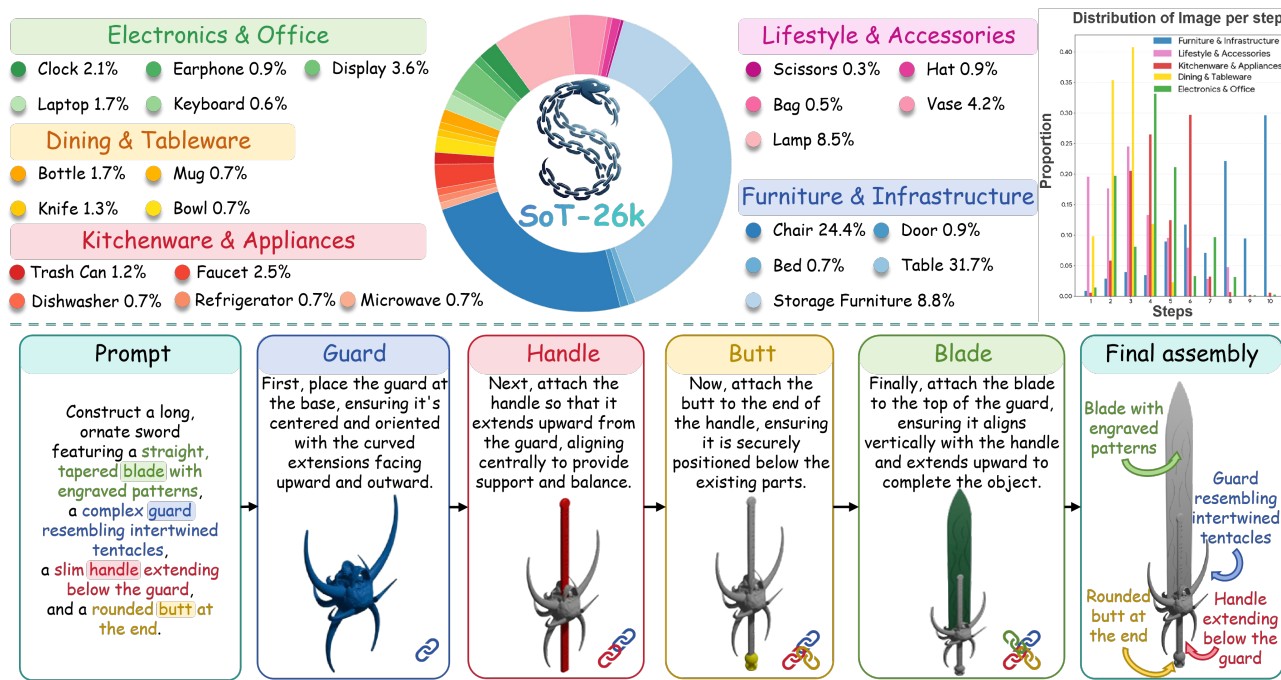

*Figure 2.* **Overview of the SoT Ecosystem.** *Top:* Statistics of the proposed **SoT-26K** dataset, showing the distribution of object categories and assembly step lengths. The data covers a wide range of structural complexities, from simple short-horizon objects to complex long-horizon assemblies. *Bottom:* The inference workflow of the SoT framework. Conditioned on a global goal prompt, the model autoregressively generates an interleaved multimodal trace, alternating between textual rationales and rendered intermediate states.

issues by decomposing synthesis into a visible, verifiable trace with stepwise visual grounding.

To bridge this gap, we propose **Shape-of-Thought (SoT)**, a framework for **progressive shape assembly that operates in the 2D rendered domain**. Unlike tool-augmented systems that rely on external engines, SoT trains a single early-fusion multimodal Transformer to generate an interleaved *text–image* trace. At each step, the model predicts a structural operation (rationale) and immediately grounds it by generating the resulting rendered state. This allows the model to "see" its own progress, providing stepwise visual feedback that promotes visual consistency.

To enable this paradigm, we construct a pipeline converting PartNet CAD assets (Mo et al., 2019) into step-aligned multimodal supervision. This yields **SoT-26K**, a large-scale dataset of 26K interleaved assembly traces, and supports **T2S-CompBench**, a benchmark that covers key aspects for evaluating structural integrity and trace faithfulness.

As shown in Figure 2, our contributions are three-fold:

- **Shape-of-Thought (SoT):** A visual Chain-of-Thought framework that decomposes shape assembly into sequential 2D visual sub-goals, enabling a single autoregressive model to capture structure-aware compositional regularities induced by part-based CAD supervi-

sion, without explicit 3D representations.

- **SoT-26K & T2S-CompBench:** A large-scale dataset of 26K grounded assembly traces with an automated generation pipeline, paired with a hybrid benchmark combining VLM semantic judging and geometric mask stability for evaluating progressive shape assembly.

- **Empirical Validation:** Comprehensive experiments showing that explicit visual grounding improves rendered-domain structural control, including +24% component numeracy and +19% structural topology over direct generation baselines.

## 2. Related Work

**From Direct Generation to Interleaved Multimodal Reasoning.** Text-to-image diffusion models can generate high-quality images from a single prompt (Ho et al., 2020; Dosovitskiy et al., 2021; Ramesh et al., 2022; Saharia et al., 2022; Rombach et al., 2022), yet often struggle with fine-grained compositional constraints, especially attribute binding and spatial relations. In language modeling, Chain-of-Thought prompting suggests that eliciting intermediate reasoning steps can improve compositional generalization (Wei et al., 2022; Lyu et al., 2023; Wang & Zhou, 2024). Recent multimodal extensions similarly externalize intermediate *visual*

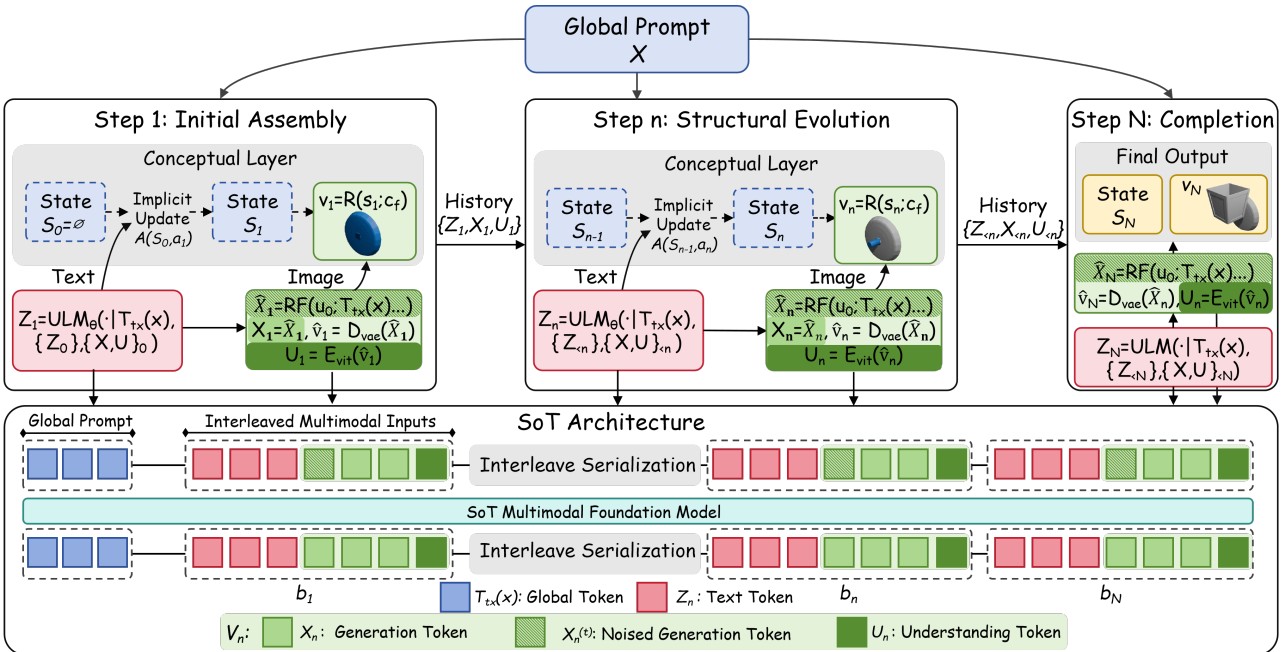

*Figure 3.* **Overview of the SoT framework.** The architecture progressively evolves from *Initial Assembly* to *Structural Evolution* and *Completion*. At each step $n$, the model utilizes a **Conceptual Layer** where textual rationales ($z_n$) serve as a scaffold to guide the generation of grounded visual tokens ($v_n$). The bottom panel illustrates the tokenization implementation, where interleaved text and vision tokens are processed by a unified multimodal foundation model.

*thoughts*, generating images as part of the reasoning trace to support spatial working memory (Chen et al., 2024; Cheng et al., 2025; Gu et al., 2025; Tang et al., 2023; Zhong et al., 2025; Zhao et al., 2025; Dong et al., 2025; Tian et al., 2025; Li et al., 2025; Zhang et al., 2025; Wang et al., 2025; Cui et al., 2025). In parallel, models based on early fusion interleave text and image tokens within a single autoregressive stream, making mixed-modal traces a native output format. (Yu et al., 2023; Team, 2025; Cao et al., 2026). Building on this paradigm, we study *process-supervised generation*, where the model outputs an interleaved *text–image* trace of intermediate assembly states, enabling stepwise supervision beyond direct generated images.

**Compositional Benchmarks and Assets.** Benchmarks such as T2I-CompBench++ provide controlled prompts to diagnose compositional failures in text-to-image generation, including attribute binding, spatial relations, numeracy, and complex compositions (Huang et al., 2025). Separately, PartNet offers large scale 3D objects with fine-grained hierarchical part annotations (Mo et al., 2019). However, compositional T2I benchmarks typically supervise only the final image, while part-based 3D assets lack language-aligned, stepwise 2D traces suitable for training process-aware generators. Prior work leverages 3D either at inference time via tool-augmented systems (Yuan et al., 2024; Yamada et al., 2025; Tang et al., 2025) or by injecting 3D representations for grounded reasoning (Hong et al., 2023; Linghu et al.,

2025; Cha & Kim, 2025). By contrast, we use part-based 3D structure to *synthesize step-aligned supervision in 2D image space*, pairing rendered intermediate assembly states with textual rationales.

## 3. Methodology

### 3.1. Shape-of-Thought

**Problem formulation.** Given a goal prompt $x$ describing a target object, Shape-of-Thought (SoT) reframes text-conditioned shape generation from direct appearance synthesis into *progressive assembly*. Rather than producing a final image directly, the model generates an interpretable multimodal trace that alternates between (i) a textual decision describing an incremental construction action and (ii) a rendered intermediate state that grounds this decision and enables stepwise visual checking. Figure 3 provides an overview of this interleaved reasoning-and-rendering framework. At each step $n \in \{1, \ldots, N\}$, where $N$ is the number of assembly steps, the model outputs a textual plan $z_n$ followed by a rendered observation $v_n$. **Formally, we treat shape assembly as a sequence of view-consistent 2D generation steps.**

**State and rendering.** Let $s_n$ denote the abstract assembly state after $n$ steps, with $s_0 = \varnothing$. In our dataset construction, $s_n$ is realized by composing part-based CAD assets

according to a deterministic schedule (fine-grained assembly scheduling; Sec. 3.2). Crucially, SoT does *not* require predicting explicit 3D geometry during generation: the model never observes meshes, point clouds, or part poses. We use $s_n$ only for trace construction and evaluation, and never expose it to the model. The model only observes the state through a fixed 2D rendering function:

$$v_n = \mathcal{R}(s_n; \mathbf{c}_f), \tag{1}$$

where $\mathcal{R}$ is our Blender rendering protocol and $\mathbf{c}_f$ is a canonical **front** camera. Using a canonical view reduces viewpoint ambiguity and stabilizes the tokenized sequence length. Our data additionally includes auxiliary renderings under fixed **left/right/back** cameras $\{\mathbf{c}_l, \mathbf{c}_r, \mathbf{c}_b\}$ to support multi-view evaluation, while the main training and evaluation protocol uses the canonical front view. Under this formulation, $s_n$ serves as the *abstract structural ground truth*, while $v_n$ provides the necessary *geometric evidence* for the model.

**Interleaved supervision.** Each example provides a step-aligned interleaved trace

$$\mathcal{T} = \{(z_1, v_1), (z_2, v_2), \ldots, (z_N, v_N)\}, \tag{2}$$

where $z_n$ specifies the incremental action at step $n$ (e.g., which part is added and how it attaches to the existing structure), and $v_n$ is the rendered consequence of that action. Conceptually, the underlying assembly evolves via an atomic action $a_n$ (part selection with placement/attachment):

$$s_n = \mathcal{A}(s_{n-1}, a_n), \qquad s_0 = \varnothing, \tag{3}$$

where $\mathcal{A}$ is the deterministic assembly operator used only during data generation. This coupling yields **mutual disambiguation**: language resolves *what* to construct and why, whereas vision resolves *where/how* the new component fits, providing explicit grounding that text-only CoT lacks.

**Token-level formulation.** We serialize each example into a unified token stream that interleaves *text tokens* and *image-token blocks*. Let $T_{tx}(\cdot)$ be the text tokenizer. Following the Bagel design (Deng et al., 2025), each rendered image $v_n$ is represented by an image-token block that bundles (i) *understanding tokens* from a ViT encoder and (ii) *generation tokens* in a frozen VAE (Kingma & Welling, 2013) latent space. During training, we also sample a continuous rectified-flow time and construct the noised latents along the linear path from data to noise, where $t_n = 0$ corresponds to

clean data and $t_n = 1$ to pure noise. Concretely,

$$Z_n = T_{tx}(z_n), \tag{4}$$
$$U_n = E_{vit}(v_n), \tag{5}$$
$$X_n = E_{vae}(v_n), \tag{6}$$
$$\tilde{X}_n^{(t_n)} = (1 - t_n)X_n + t_n\epsilon_n, \tag{7}$$
$$t_n \sim \mathcal{U}(0, 1), \quad \epsilon_n \sim \mathcal{N}(0, I), \tag{8}$$

where $E_{vit}$ denotes the vision encoder for understanding tokens, and $E_{vae}$ is a *frozen* VAE encoder. We treat the concatenation for training-time

$$V_n = T_i(v_n; t_n, \epsilon_n) \triangleq \left[\tilde{X}_n^{(t_n)}, X_n, U_n\right] \tag{9}$$

as the image-token block for step $n$, where $\epsilon_n$ is sampled noise. Within each image block, we order visual tokens as (noised VAE, clean VAE, ViT). We then build the per-step block

$$\mathbf{b}_n = (Z_n, V_n), \tag{10}$$

and the full training sequence

$$\mathbf{y} = (T_{tx}(x), \mathbf{b}_1, \ldots, \mathbf{b}_N), \tag{11}$$

where special tokens delimit modalities via reserved textual tokens.

**Unified interleaved decoding.** Let $ULM_\theta$ denote a unified multimodal decoder that (i) defines an autoregressive conditional distribution over text tokens given an interleaved prefix and (ii) implements a rectified-flow velocity predictor over VAE latents. SoT follows a *decide-then-ground* order: at step $n$, the model first generates the textual plan $Z_n$ and then generates the rendered observation by rectified-flow sampling in the VAE latent space:

$$Z_n \sim ULM_\theta\Big(\cdot \mid T_{tx}(x), \{Z_j\}_{j<n}, \{X_j, U_j\}_{j<n}\Big), \tag{12}$$

$$\hat{X}_n \leftarrow \text{RF-Sample}\Big(u_\theta; T_{tx}(x), \{Z_j\}_{j\leq n}, \{X_j, U_j\}_{j<n}\Big), \tag{13}$$

$$\hat{v}_n = D_{vae}(\hat{X}_n), \tag{14}$$

where $u_\theta$ is the rectified-flow velocity field predicted by $ULM_\theta$ and queried during sampling, and $D_{vae}$ is the *frozen* VAE decoder. After obtaining $\hat{v}_n$, we set $X_n = \hat{X}_n$ and compute $U_n = E_{vit}(\hat{v}_n)$ as deterministic conditioning tokens for subsequent steps (Appendix B). Importantly, $X_n$ is produced via rectified-flow sampling in the continuous VAE latent space, and $U_n$ is deterministically encoded from $\hat{v}_n$; the model does not sample them as discrete codebook tokens.

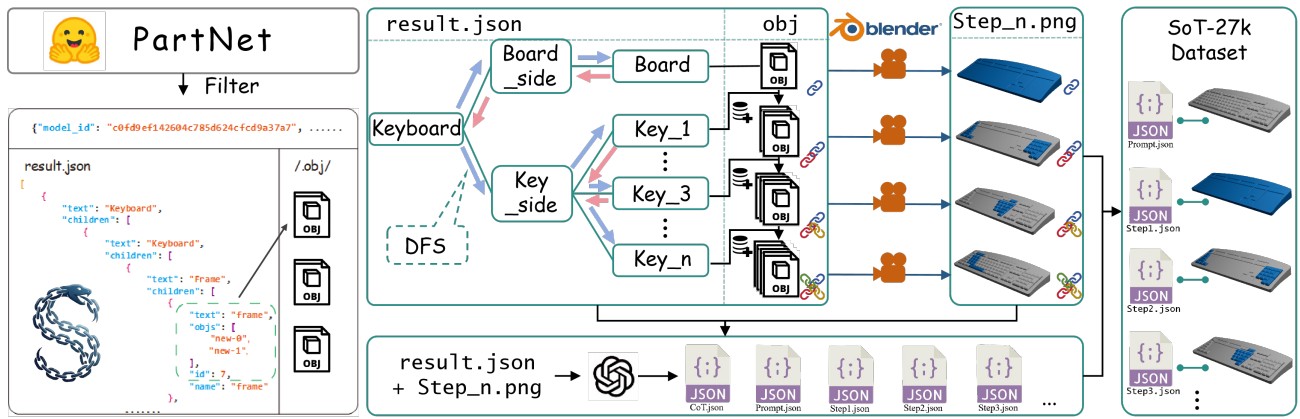

*Figure 4.* **The SoT-26K Construction Pipeline.** We transform PartNet assets into multimodal traces through four stages: (A) **Data Curation** to filter and validate raw hierarchies; (B) **Hierarchy Decomposition** to enforce fine-grained assembly schedules; (C) **Automated Rendering** to generate cumulative intermediate states in Blender; and (D) **Multimodal Annotation**, where GPT-4o synthesizes stepwise rationales and goal prompts grounded in the visual evidence.

## 3.2. SoT-26K Dataset

**Why SoT-26K?**  Progressive assembly requires supervision beyond goal conditioning: the model must learn *how* structure evolves, step by step, with each textual decision grounded by a corresponding visual state. We therefore curate SoT-26K, a large-scale collection of *step-aligned* interleaved state–rationale traces. Two design principles make supervision more unambiguous. First, we enforce a fine-grained schedule where each step introduces a small batch of leaf parts that together form one semantic component, capped by a maximum batch size (e.g., 15–25 leaf parts per step for complex objects), yielding a uniquely defined structural delta $\Delta P_n$. Second, we constrain each rationale to a lightweight slot schema—(*new component*) + (*attachment relation*) + (*anchor on existing structure*)—to promote semantic consistency and discourage free-form hallucinations.

**Dataset overview.**  SoT-26K contains **25,929** object traces (denoted as 26K for brevity) across **24** diverse object categories, each paired with a single goal prompt $x$ describing the target object. Although derived from part-based 3D CAD assets dataset PartNet (Mo et al., 2019), SoT-26K is packaged as a vision–language dataset: all supervision is delivered via *2D renderings* produced by a fixed Blender protocol under a canonical front camera. Auxiliary left/right/back renders support multi-view evaluation and analysis.

**Interleaved trace definition.**  Each instance is represented as a progressive assembly trace

$$\mathcal{T} = \big[x, (z_1, v_1), (z_2, v_2), \ldots, (z_N, v_N)\big], \quad (15)$$

where $x$ is the goal prompt, $v_n$ is the rendered image of the cumulative assembly at step $n$, and $z_n$ is a concise rationale grounded in the transition from step $n-1$ to $n$. Let $P_{\leq n}$ denote the cumulative set of assembled parts at step $n$ (with $P_{\leq 0} = \varnothing$). Under our fine-grained assembly schedule, the newly added parts at step $n$ are uniquely defined as

$$\Delta P_n = P_{\leq n} \setminus P_{\leq n-1}. \quad (16)$$

We generate $z_n$ to explicitly describe $\Delta P_n$ and its attachment to an existing anchor in $P_{\leq n-1}$, while $v_n$ provides visual evidence of the updated structure.

**Construction pipeline.**  We construct SoT-26K from PartNet-style assets through four stages: **(A) Data curation and loading** preprocesses the raw PartNet dataset by removing duplicates and organizing samples by category, then loads and validates JSON hierarchies. **(B) Hierarchy decomposition** parses the part tree and produces an ordered leaf-part schedule $\{P_{\leq n}\}_{n=1}^{N}$, enforcing a fine-grained assembly schedule (Eq. 16). **(C) Automated rendering** generates the cumulative intermediate states $\{v_n\}$ in headless Blender under a canonical front camera. **(D) Multimodal annotation** uses an MLLM (GPT-4o) to generate the goal prompt $x$ from the final rendering and to produce step rationales $\{z_n\}$ grounded in $\Delta P_n$ and conditioned on rendered visual evidence. Full implementation details, file formats, and prompt templates are provided in Appendix A.2.

## 3.3. Model and Training

**Instantiation.**  We instantiate $\text{ULM}_\theta$ in Sec. 3.1 with Bagel-7B (MoT variant), a unified multimodal autoregressive Transformer (Deng et al., 2025). The MoT architecture employs hard routing where VAE generation tokens are processed by a dedicated generation expert, while text tokens

and ViT understanding tokens are handled by the understanding expert. Given the serialized interleaved stream $\mathbf{y}$ (Eq. 11), Bagel performs early-fusion modeling by attending jointly over textual tokens (plans and delimiters) and interleaved image blocks composed of continuous VAE latent tokens (for generation) and ViT understanding tokens (for perception). Here, 'token' refers to an element in the multimodal sequence: text tokens are discrete, while visual-latent tokens are continuous VAE latents projected into the model's embedding space. This design enables tight coupling between structural planning and visual grounding.

**Training and inference.** We train with teacher forcing on the serialized interleaved stream $\mathbf{y}$ (Eq. 11). Text tokens are optimized with standard next-token cross-entropy, while visual segments are optimized by regressing the target visual latents produced by the *frozen* pretrained image tokenizer/VAE.

We fine-tune the Bagel-7B Transformer backbone jointly with the ViT encoder, while keeping the pretrained VAE tokenizer (encoder/decoder) frozen. Optimization follows Table 4 (AdamW, cosine LR with warmup, gradient clipping, EMA, token-budgeted batching). At inference time, the model decodes the interleaved stream in the *decide-then-ground* order (Eqs. 12–14) until the final visual segment $V_N$ is produced; decoding $V_N$ through the fixed VAE yields the final rendered output. Additional training and decoding details are provided in Appendix B.

### 3.4. Evaluation Benchmark: T2S-CompBench

Appearance-centric metrics (Hessel et al., 2021; Huang et al., 2025) do not assess whether an object is constructed through a faithful and stable step-by-step trace. We therefore introduce **T2S-CompBench (Text-to-Shape)**, a benchmark that captures process information along two axes: **T2S-Structure** evaluates whether the final rendering $v_N$ satisfies the goal prompt $x$, and **T2S-Process** evaluates whether the intermediate trace $\{v_1, \ldots, v_N\}$ is stable and step-faithful. T2S-Structure reports five forced-choice metrics: **CN** (Component Integrity and Numeracy), **SF** (Shape Fidelity), **AF** (Attribute Fidelity), **CP** (Connectivity Plausibility), and **VT** (Visual Topology). T2S-Process reports two trace metrics: **TS** (Trace Stability) and **RA** (Stepwise Rationale Alignment). Details and formulas are shown in Appendix E.

We adopt a hybrid evaluation protocol tailored to our rendered domain: GPT-4o (Hurst et al., 2024) (API) provides closed-form forced-choice decisions for semantic/relational metrics ({CN,SF,AF,CP,VT,RA}), while SAM 3 (Carion et al., 2025) union foreground masks are used to compute TS. Unless otherwise noted, all metrics are computed on the canonical **front** view. To validate robustness against potential VLM-as-a-judge bias, we re-audit a stratified subset with

an alternative open-weight VLM (Qwen2-VL-7B-Instruct (Wang et al., 2024)) and quantify cross-judge consistency in Appendix I.

### 3.5. Human Evaluation

We conduct a human perceptual study on the final synthesized results to complement automated metrics. Each example is rated on a **5-point Likert scale** along three aspects: (1) **Visual Quality** (fidelity and details), (2) **Structural Correctness** (topology and completeness), and (3) **Construction Logic** (whether the final object exhibits a physically plausible assembly structure). We report the mean score averaged over the three aspects. The full study protocol (sampling, blinding/randomization, aggregation, confidence intervals, and agreement) is provided in Appendix H.

## 4. Experiments

**Data.** We use SoT-26K (Sec. 3.2), consisting of 25,929 object traces derived from 3D CAD assets across 24 object categories, each containing interleaved *text–image* assembly traces. We split the annotated data into 69.9%/10.1%/20.0% for train/val/test following the original PartNet split (Mo et al., 2019), resulting in 18,141 training samples, 2,619 validation samples, and 5,169 test samples. Unless otherwise specified, we train and evaluate using the canonical **front** view, and truncate each serialized sequence to at most 50K tokens under token-budgeted batching with a soft target of 40K tokens per batch.

**Model.** We fully fine-tune BAGEL-7B (MoT variant) (Deng et al., 2025) end-to-end for interleaved text and visual-latent generation. Specifically, we perform full-parameter fine-tuning of the language model and vision transformer backbone modules, while keeping the VAE encoder/decoder frozen. All experiments are conducted on $32\times$NVIDIA H100 (80GB) GPUs, with a total training time of 16.1 hours.

**Optimization.** We train for 2,000 steps with AdamW ($\beta_1$=0.9, $\beta_2$=0.95, $\epsilon$=$10^{-8}$), a cosine learning-rate schedule with 50 warmup steps, peak learning rate $2 \times 10^{-5}$, and minimum learning rate $10^{-6}$. We clip gradients to a global norm of 1.0 and use EMA with decay 0.9999. We cap the token budget by setting the expected number of tokens per step to 40K and the maximum to 50K (with max 50K per sample). Additional hyperparameters are deferred to Appendix D.

### 4.1. Results

**Main results on T2S-CompBench.** Table 1 compares SoT with (i) direct generation and text-only CoT 2D generation on the same Bagel-7B backbone, and (ii) representative 3D synthesis / tool-driven systems (Shap-E, LGM, Meshy 6)

*Table 1.* **Quantitative Results on T2S-CompBench.** We compare SoT against 2D generative baselines and 3D synthesis methods. Note that **Meshy 6** is included as a *commercial reference* for strong appearance quality. **SoT** achieves substantially stronger structural compliance on Numeracy (CN) and Topology (VT) in the projected views. The best and second-best scores are highlighted. The last column reports the average **Human Evaluation** score (1–5 scale) across Visual Quality, Structural Correctness, and Construction Logic. RA/TS require visual traces while some baselines only provide final assets (–).

| Method | T2S-Structure | | | | | T2S-Process | | Human (1–5)↑ | Latency / s |
|---|---|---|---|---|---|---|---|---|---|
| | CN↑ | SF↑ | AF↑ | CP↑ | VT↑ | RA↑ | TS↑ | | |
| *2D Generative Baselines* | | | | | | | | | |
| Bagel-7B | 64.26 | 71.57 | 58.34 | 62.14 | 65.42 | – | – | 3.12 | 51.95 |
| Bagel-7B-CoT[†] | 75.88 | 74.23 | 72.16 | 68.92 | 71.38 | 45.49 | 32.71 | 3.65 | 103.46 |
| *Rendered 3D baselines[‡]* | | | | | | | | | |
| Shap-E (Jun & Nichol, 2023) | 42.15 | 75.38 | 25.23 | 21.11 | 28.59 | – | – | 1.85 | 9.92 |
| LGM (Tang et al., 2025) | 68.62 | 80.15 | 55.40 | 72.30 | 76.50 | – | – | 3.25 | 6.48 |
| L3GO (Yamada et al., 2025) | 76.20 | 65.80 | 68.45 | 60.10 | 72.90 | – | – | 3.05 | 921.27[◇] |
| Meshy 6[*] | 82.74 | 95.43 | 75.27 | 85.60 | 78.25 | – | – | 3.91 | 74.81 |
| **Bagel-7B-SoT** | 88.44 | 83.62 | 81.51 | 86.25 | 84.76 | 79.19 | 91.30 | 4.08 | 43.14 (per step) / 257.75 (total) |

[†] For *Bagel-7B-CoT*, we generate the full textual CoT first, then sample corresponding images. Step-controlled CoT-K and Refine-K comparisons are reported in Appendix F.
[‡] We render 3D baselines' outputs with the same canonical camera protocol.
[◇] L3GO relies on iterative trial-and-error loops within the Blender environment.
[*] Meshy 6 is a closed-source commercial service evaluated via the Meshy 6 API (no-texture mode). https://www.meshy.ai/blog/meshy-6-launch.

whose outputs are evaluated via our canonical rendering protocol. SoT achieves the best performance on the majority of **T2S-Structure** metrics, demonstrating superior compliance in part counting (CN), attributes (AF), and spatial topology (VT). Compared to **CoT** sampling with text alone, SoT yields especially large gains on assembly-related constraints, such as attachment and spatial relations, while maintaining high fidelity on attributes tied to parts. A budget-controlled comparison in Appendix F further shows that these gains are not explained by additional decoding alone: CoT-K reaches 72.49 Avg. at 79.82s, Refine-K reaches 73.67 Avg. at 248.97s, while SoT reaches 85.09 Avg. at 259.13s.

SoT additionally enables **T2S-Process** evaluation by generating explicit intermediate states. Interleaving rendered states substantially stabilizes the generation trajectory: TS improves from 32.71 to 91.30, and stepwise rationale alignment reaches 79.19. These results suggest that supervising decide-then-ground traces helps the model maintain a consistent visual working memory over multi-step assembly, reducing drift and uncontrolled re-drawing across steps.

**Rendered-view structural projection comparison.** Native 3D systems are included as contextual references rather than direct baselines for SoT, which produces rendered states rather than editable 3D assets. We render their outputs into canonical views to ask whether high-fidelity 3D generators already satisfy the same fine-grained compositional constraints under our view-based protocol. This comparison evaluates rendered structural compliance under a shared view-based protocol.

Shap-E and LGM (Jun & Nichol, 2023; Tang et al., 2025) attain reasonable Shape Fidelity (SF), but they struggle with fine-grained binding and topology (AF/CP). Meshy 6 achieves strong global connectivity (SF/CP) yet falls short on strict compositional constraints, specifically Numeracy (CN) and Attributes (AF) due to hallucinations and detail omission. Appendix G additionally reports T2S-CompBench-MV results, while Appendix J discusses multi-view failure cases that clarify where the current single-view SoT setting remains limited.

**Visualization.** Figure 5 visualizes the complete inference traces. Unlike black-box generation, SoT explicitly decomposes high-level goals into sequential visual sub-tasks, alternating between textual rationales and grounded visual states. For instance, in the **S-shaped chair** example (3rd column), the model maintains global geometric consistency, making the backrest connect seamlessly to the base rather than floating. This suggests that intermediate renderings serve as a **visual working memory**, facilitating the model to self-correct and verify structural integrity step-by-step.

**Qualitative Analysis: Fine-Grained Structural Compliance.** Figure 6 provides a fine-grained qualitative analysis of structural compliance under T2S-CompBench constraints. Across electronics, lamps, appliances, faucets, and earphones, the compared baselines exhibit distinct failure modes: 2D generative baselines may preserve the coarse object category but drift on repeated components, local attributes, or fine-grained layouts, while rendered 3D baselines often produce plausible global silhouettes but collapse thin structures, smooth out task-critical details, or disconnect/misplace functional parts. These cases show that visual plausibility alone does not guarantee structural compliance. In contrast, SoT uses intermediate visual states as explicit

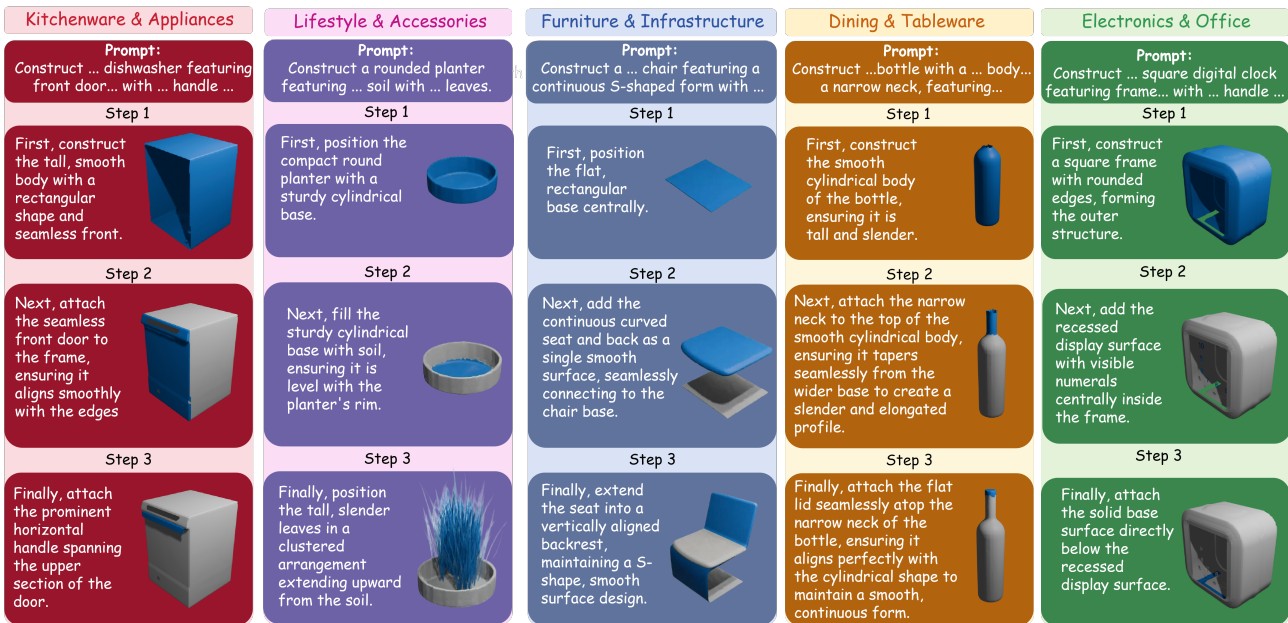

*Figure 5.* **Progressive Shape Assembly Traces.** SoT decomposes each goal prompt into sequential construction steps across diverse object categories. At each step, the model produces a structural rationale followed by a grounded visual state, making the generation process explicit and visually traceable.

grounding, helping preserve component counts, topology, attributes, and part connectivity throughout the generation trajectory.

## 4.2. Ablation Study

Table 2 analyzes the impact of mechanism and data choices. Regarding mechanism, removing interleaved visual states (*No Visual Thoughts*) degrades structural logic (CP drops from 86.25 to 68.92), while masking visual history (*w/o Visual History*) notably impacts stability (TS decreases to 42.83), suggesting the role of visual working memory. In terms of data, replacing the structured schema with free-form text (*Free-form Rationale*) reduces Rationale Alignment (RA drops to 55.47) while maintaining similar structural scores, indicating that the schema specifically aids in aligning semantic plans with visual updates.

Together with the budget-controlled baselines in Appendix F, these ablations rule out a simple budget explanation: CoT-K/Refine-K add reasoning or re-decoding budget yet trail SoT, and removing visual states/history lowers CP or TS under the same evaluation. The main benefit therefore comes from coupling each construction decision with a visible intermediate state. Removing visual thoughts collapses the method toward text-only planning, while masking visual history shows that later steps benefit from prior rendered states rather than independent re-sampling. Finally, the free-form rationale variant preserves much of the final structural quality but substantially reduces rationale alignment, indi-

*Table 2.* **Ablation Study.** We validate key design choices across mechanism, data, and optimization. **Full Model** is Bagel-7B (SoT). *No Visual Thoughts* removes interleaved image generation. *w/o Visual History* masks attention to prior visual tokens. *Free-form Rationale* removes the structured text schema.

| Variant | Structure | | | | Process | |
|---|---|---|---|---|---|---|
| | CN ↑ | AF ↑ | CP ↑ | VT ↑ | RA ↑ | TS ↑ |
| **SoT (Full Model)** | **88.44** | **81.51** | **86.25** | **84.76** | **79.19** | **91.30** |
| *A. Mechanism* | | | | | | |
| No Visual Thoughts | 75.88 | 72.16 | 68.92 | 71.38 | 45.49 | 32.71 |
| w/o Visual History | 84.32 | 78.17 | 65.44 | 67.51 | 71.16 | 42.83 |
| *B. Data Construction* | | | | | | |
| Free-form Rationale | 86.59 | 79.80 | 83.13 | 82.94 | 55.47 | 88.25 |

cating that the lightweight slot schema mainly improves the faithfulness and inspectability of the generated process.

## 5. Conclusion

We introduced **Shape-of-Thought (SoT)**, a framework reframing rendered shape generation as progressive assembly via interleaved text-image traces. To facilitate this paradigm, we curated the **SoT-26K** dataset and established **T2S-CompBench** for rigorous structural evaluation. This approach enables a single multimodal model to perform stepwise generation in image space without external 3D engines at inference time. Our results validate that intermediate visual states serve as an effective reasoning substrate

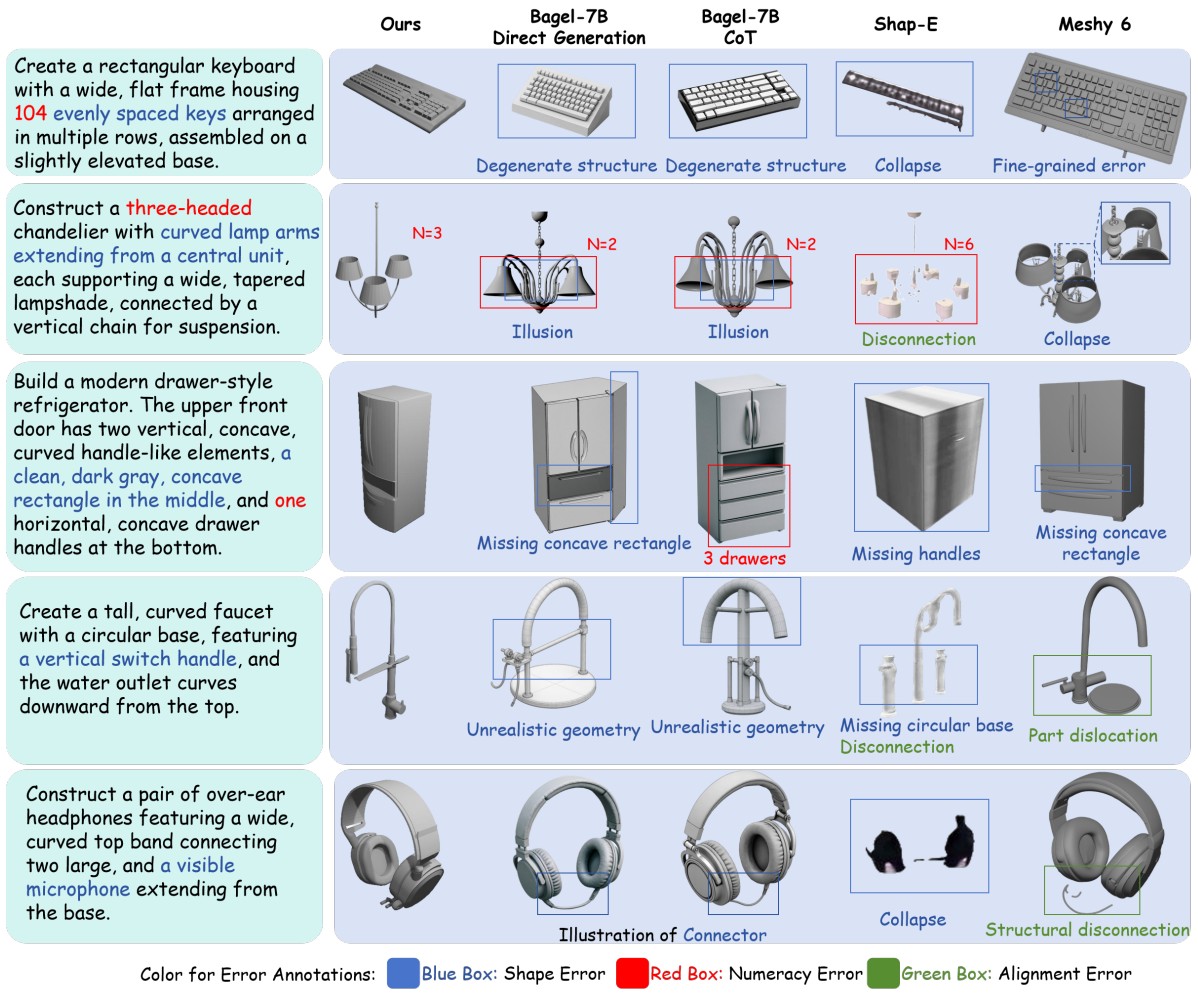

*Figure 6.* **Qualitative Comparison on Fine-Grained Structural Compliance.** We compare SoT with representative 2D and rendered-3D baselines across diverse object categories. The examples cover structural failures beyond numeracy, including degenerate or collapsed geometry, missing fine-grained attributes, misplaced parts, and disconnected components. SoT better preserves component counts, object-level topology, local details, and part connectivity. Blue, red, and green boxes mark shape/detail failures, count mismatches, and connectivity/dislocation errors, respectively.

for structural composition under rendered supervision.

**Discussion.** Building on SoT, future research may consider several important directions:

1. **From SoT renders to textured 3D assets.** A natural extension is to use SoT renderings as structured image inputs for mask-to-3D or image-to-3D pipelines (Team et al., 2025). Appendix K illustrates this downstream use qualitatively; systematic 3D asset evaluation requires dedicated metrics and baselines.

2. **Efficient training for multi-view.** Develop compute-aware strategies to unlock multi-view supervision, which is currently constrained by the prohibitive computational costs of dense visual traces (already saturating 32×H100 GPUs). We provide a detailed limitation and mitigation directions in Appendix J.

3. **Physical interpretability beyond transparent traces.** Although SoT exposes intermediate rationales and states, the learned procedure is not yet interpretable in terms of physical constraints, limiting faithful "assembly" understanding. A practical next step is to augment traces with lightweight checks for contact, support, symmetry, and part persistence, using rendered masks or inferred part correspondences as diagnostics rather than external engines at inference time. Such checks would clarify whether an intermediate state represents a valid structural update or only a visually plausible redraw, and would provide more actionable feedback when topology or connectivity fails.

## Impact Statement

**Broader Impact.** This paper introduces Shape-of-Thought (SoT), a framework that reframes rendered structural object generation as a progressive, interpretable visual reasoning process. By shifting from opaque, black-box synthesis to a transparent, step-by-step assembly paradigm, our work contributes to the development of more controllable and explainable AI systems.

**Interpretability and User Control.** A key societal benefit of SoT is the enhancement of transparency in generative AI. Traditional image or 3D generation systems often function as "black boxes," making it difficult for users to understand why a specific structure was generated or to correct errors. SoT's visual chain-of-thought allows users to inspect the intermediate rationale and visual states. This interpretability is crucial for professional applications where verifying structural logic is as important as final appearance.

**Ethical Considerations and Risks.** We acknowledge potential risks associated with generative models. Like all models trained on large-scale datasets, SoT may inherit biases present in the training data (Mo et al., 2019), potentially reflecting cultural stereotypes in object design. Furthermore, while the focus of this work is on inanimate objects, the technology could theoretically be adapted to generate harmful content. We have focused our dataset on safe, common object categories, but future deployment should include rigorous safety filters and content moderation mechanisms.

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

# A. SoT-26K Details

## A.1. SoT-26K Statistics

Table 3 provides detailed statistics of SoT-26K, including category breakdown, assembly complexity, and step distributions.

The dataset encompasses 25,929 object traces derived from 3D CAD assets across 24 object categories, with each asset converted into an assembly trace averaging 6.2 steps (range 2-12). Categories are organized by functional groups, with furniture dominating due to structural complexity. The step distribution reveals assembly complexity patterns: furniture objects average 7.5 steps, appliances 5.8 steps, and tools 4.2 steps.

*Table 3.* Detailed Statistics of SoT Dataset by Category

| General Category | Sub Category | Count | Avg Steps | Percentage (%) |
|---|---|---|---|---|
| Furniture & Infrastructure | Table | 8,227 | 7.6 | 31.7 |
| | Chair | 6,323 | 7.5 | 24.4 |
| | StorageFurniture | 2,269 | 8.1 | 8.8 |
| | Door | 230 | 2.9 | 0.9 |
| | Bed | 192 | 8.4 | 0.7 |
| | **Subtotal** | **17,241** | **7.5** | **66.5** |
| Lifestyle & Accessories | Lamp | 2,212 | 4.2 | 8.5 |
| | Vase | 1,086 | 1.8 | 4.2 |
| | Hat | 231 | 2.1 | 0.9 |
| | Bag | 126 | 2.6 | 0.5 |
| | Scissors | 68 | 4.8 | 0.3 |
| | **Subtotal** | **3,723** | **3.4** | **14.4** |
| Electronics & Office | Display | 930 | 3.6 | 3.6 |
| | Clock | 554 | 2.9 | 2.1 |
| | Laptop | 433 | 4.9 | 1.7 |
| | Earphone | 229 | 6.4 | 0.9 |
| | Keyboard | 156 | 7.0 | 0.6 |
| | **Subtotal** | **2,302** | **4.1** | **8.9** |
| Kitchenware & Appliances | Faucet | 648 | 4.4 | 2.5 |
| | TrashCan | 321 | 4.6 | 1.2 |
| | Refrigerator | 187 | 5.4 | 0.7 |
| | Microwave | 183 | 4.5 | 0.7 |
| | Dishwasher | 181 | 4.1 | 0.7 |
| | **Subtotal** | **1,520** | **4.5** | **5.9** |
| Dining & Tableware | Bottle | 436 | 2.8 | 1.7 |
| | Knife | 327 | 3.3 | 1.3 |
| | Mug | 192 | 2.3 | 0.7 |
| | Bowl | 188 | 1.5 | 0.7 |
| | **Subtotal** | **1,143** | **2.6** | **4.4** |
| **Total** | | **25,929** | **6.2** | **100.0** |

**Step Distribution Analysis:** The dataset exhibits diverse assembly complexities across categories. Furniture objects show the highest step counts (7.5 average), reflecting their hierarchical part structures and multi-component assemblies. Dining objects have simpler assemblies (2.6 average), often consisting of basic geometric components. The step distribution follows a right-skewed pattern, with most objects (85%) requiring 4-9 assembly steps.

## A.2. SoT-26K Construction Details

**Overview.** This appendix provides additional details on the construction of SoT-26K, our large-scale dataset for progressive part-based assembly traces. Starting from raw PartNet (Mo et al., 2019) CAD assets, we transform hierarchical 3D part structures into step-aligned multimodal traces suitable for training multimodal autoregressive models. The construction pipeline addresses several key challenges: (i) maintaining structural consistency across assembly steps, (ii) maintaining visual coherence in rendered intermediate states, (iii) generating semantically meaningful assembly prompts, and (iv) scaling the process to handle thousands of diverse objects.

Our dataset encompasses **25,929** object traces derived from 3D CAD assets spanning **24** diverse object categories, from furniture (tables, chairs, storage units) to appliances (refrigerators, microwaves) to everyday objects (bottles, scissors, bags). Each asset is converted into an assembly trace with an average of 6.2 steps, resulting in **162,003** step-level annotations across all traces. Each trace consists of interleaved text–image pairs, where textual rationales describe incremental assembly decisions and images provide visual grounding of the resulting structural state.

The construction process is organized into four interdependent stages that progressively transform raw CAD data into training-ready multimodal sequences: **(A) Data curation and loading** helps ensure dataset quality and structural validity; **(B) Hierarchy decomposition** creates assembly schedules with controlled granularity; **(C) Automated rendering** produces consistent visual representations of intermediate states; **(D) Multimodal annotation** generates natural language descriptions aligned with visual changes.

Each stage incorporates quality control measures and handles edge cases specific to CAD data processing. The resulting dataset enables supervised learning of assembly logic, with each trace providing explicit step-by-step supervision for multimodal generation.

**Stage A: Data curation and loading.** **Input:** raw PartNet dataset with scattered sample folders containing `meta.json`, `result.json`, and `objs/` directories.
**Output:** validated part hierarchies organized by category with quality metrics.

The curation phase begins with systematic scanning of all sample directories to extract metadata from `meta.json` files. Each file contains essential identifiers: `model_id` (unique model identifier), `model_cat` (semantic category like "Chair" or "Table"), and `anno_id` (annotation version identifier). We perform deduplication by `model_id` to eliminate redundant instances that may arise from multiple annotation versions, to ensure each unique 3D model appears exactly once in our dataset. Samples are then organized by category to enable category-specific processing parameters in downstream stages.

For structural validation, we load each `result.json` file and traverse the part hierarchy using depth-first search. The hierarchy represents objects as trees where internal nodes denote part groupings and leaf nodes contain mesh references in the `objs/` directory. We extract all leaf nodes with valid mesh references, recording their semantic names, textual descriptions, and OBJ file paths. Validation checks include: (i) existence of referenced mesh files, (ii) non-empty part hierarchies, (iii) consistent naming conventions, and (iv) absence of degenerate geometries.

Quality control measures during this stage include logging of parsing failures, statistics on hierarchy depths and branching factors, and identification of samples requiring manual review. This preprocessing ensures that only structurally sound, properly annotated models proceed to assembly planning, with category organization enabling efficient batch processing and parameter tuning.

**Stage B: Hierarchy parsing and assembly scheduling.** **Input:** validated part hierarchies with leaf node collections.
**Output:** assembly sequences with controlled step granularity and semantic ordering.

Assembly scheduling transforms static part hierarchies into dynamic construction sequences that simulate realistic object assembly. For each object instance $o$, we extract the complete set of leaf parts $\mathcal{L}(o) = \{p_1, \ldots, p_N\}$ through depth-first traversal of the validated hierarchy. Each leaf part $p_n$ encapsulates multiple attributes: semantic name (e.g., "seat", "backrest"), textual description, unique identifier, and associated mesh file path.

The core challenge is determining assembly order while maintaining structural plausibility and controlling sequence length. We implement a configurable assembly scheduler that balances between fine-grained (many small steps) and coarse-grained (fewer large steps) assembly. The scheduler uses category-specific parameters: furniture objects like tables and chairs use broader steps (up to 15 parts per step) to manage complexity, while precision objects like scissors and knives use finer steps

(maximum 5 parts per step) to preserve assembly precision.

Sequence ordering follows ergonomic and structural principles. Foundational parts such as bases, frames, and bodies are assembled first to establish stable foundations. Category-aware prioritization promotes object-specific assembly logic, such as assembling bottle bodies before necks or chair bases before seats. Symmetric parts like multiple legs or keyboard keys are batched together to maintain visual and structural coherence throughout the assembly process.

The resulting assembly schedule defines cumulative part sets:

$$P_{\leq n} = \{p_1, \ldots, p_n\}, \quad n = 1, \ldots, N. \tag{17}$$

This indicates that $\Delta P_n = P_{\leq n} \setminus P_{\leq n-1}$ represents a meaningful incremental addition at each step.

Sequence statistics reveal the diversity of assembly complexity: tables average 9 steps (range 6-12), chairs average 7 steps (6-10), while simple objects like mugs average 3 steps (2-6). This variation is intended to help the model learn both simple and complex assembly patterns, with quality control verifying that no sequence exceeds maximum step limits or contains implausible orders.

**Stage C: Headless rendering of intermediate assembly states.** **Input:** assembly sequences $\{P_{\leq n}\}$ with mesh file references.
**Output:** synchronized visual and metadata sequences for each assembly trace.

Rendering intermediate assembly states requires careful synchronization between 3D geometry and visual representation. For each step $n$ in the assembly sequence, we compose the cumulative part set $P_{\leq n}$ by importing all referenced OBJ meshes into a Blender scene. Meshes are scaled uniformly (3x factor) and positioned according to their original CAD coordinates, facilitating geometric consistency across assembly steps.

The rendering pipeline operates in headless mode for scalability, processing thousands of assembly states efficiently. We employ Blender's Eevee renderer with optimized settings: 512×512 resolution, 8 samples for anti-aliasing, and transparent background to isolate object geometry. A canonical front camera ($\mathbf{c}_f$) provides consistent viewpoint across all samples, reducing viewpoint variance and focusing model attention on structural changes rather than camera motion.

Scene configuration includes standardized lighting (single key light at 45° elevation) and material assignment (uniform gray Principled BSDF shader for all parts). This controlled environment maintains visual consistency while highlighting structural relationships through shading and depth cues.

For each assembly step, we generate synchronized outputs. The visual state is captured in `step_{n}.png`, showing the rendered image of the cumulative assembly up to that step. Structural metadata is stored in `step_{n}.json`, containing the step index, cumulative part list, and change description. For the complete assembly, we provide `final_complete.png` and `final_complete.json` to represent the fully assembled object.

Quality assurance includes validation of mesh loading, detection of rendering artifacts, and verification that visual changes correspond to structural additions. Memory management through periodic cleanup prevents accumulation of orphaned data blocks during batch processing. The resulting visual sequences provide pixel-level grounding for each assembly decision, enabling the model to learn the visual consequences of structural choices.

**Stage D: Text annotation for goal and step rationales.** **Input:** rendered visual sequences and structural metadata for each assembly trace.
**Output:** natural language goal prompts and step-by-step rationales synchronized with visual states.

The annotation stage bridges visual assembly states with natural language descriptions, enabling multimodal training (Cao et al., 2024). We employ GPT-4o as the annotation engine, leveraging its vision-language capabilities to generate contextually appropriate descriptions, with a total API expenditure of $1,092 for the full dataset.

**Goal prompt generation:** Each trace receives a single high-level instruction $x$ describing the complete target object. The prompt combines the final rendered image $v_N$ with the complete part list $P_{\leq N}$, instructing the model to generate imperative descriptions that capture both visual appearance and functional structure. Examples include "Build a sturdy wooden chair with four legs, cushioned seat, and curved backrest" or "Construct a modern refrigerator with double doors, stainless steel finish, and internal shelving." These instructions serve as global context for the entire assembly sequence. Prompt templates are shown in Appendix M.

**Step rationale generation:** For each incremental step $n$, we generate a precise rationale $z_n$ explaining the structural change from $v_{n-1}$ to $v_n$. The annotation follows a structured schema to improve consistency: (*action verb*) + (*new parts*) + (*attachment preposition*) + (*anchor location*). Multimodal input includes the current step image $v_n$, previous step image $v_{n-1}$ (for $n > 1$), and the computed part difference $\Delta P_n$. The model is prompted to focus on essential structural relationships while avoiding redundant explanations.

Quality control measures include consistency validation through cross-checking that mentioned parts exist in $\Delta P_n$, semantic coherence by promoting rationales use appropriate assembly terminology, length control to maintain concise descriptions (typically 10-20 words per step), and category adaptation using domain-specific vocabulary (e.g., "blade" vs "seat").

The resulting annotations create a complete multimodal trace where each visual state is precisely described by its accompanying rationale, enabling the model to learn the mapping between structural decisions and their linguistic expression. This synchronization is crucial for training autoregressive models that can generate both coherent assembly sequences and appropriate explanatory text.

**Dataset packaging and directory structure.** **Output artifact:** self-contained multimodal traces with comprehensive metadata and quality assurance.

Each processed instance is converted into a structured Parquet record that encapsulates all multimodal components necessary for training. The Parquet format promotes efficient storage and loading of interleaved text-image sequences.

The dataset is released in Parquet format for efficient storage and loading, with the following structure:

```
SoT/
+-- Bottle/
|   +-- train/
|   |   +-- train-00000-of-00001.parquet
|   +-- val/
|   |   +-- val-00000-of-00001.parquet
|   +-- test/
|       +-- test-00000-of-00001.parquet
+-- Chair/
|   +-- train/
|   |   +-- train-00000-of-00001.parquet
|   +-- val/
|   |   +-- val-00000-of-00001.parquet
|   +-- test/
|       +-- test-00000-of-00001.parquet
+-- ... (other categories)
+-- Table/
    +-- train/
    |   +-- train-00000-of-00001.parquet
    +-- val/
    |   +-- val-00000-of-00001.parquet
    +-- test/
        +-- test-00000-of-00001.parquet
```

**Parquet file schema:** Each Parquet file contains the following fields for multimodal training:

- `Prompt` (string): The high-level goal instruction describing the target object

- `Shape of Thought Reasoning Trace` (string): Formatted assembly steps with image placeholders for progressive reasoning

- `Final Assembly` (string): Fixed-format final answer indicating assembly completion

- `reasoning_image_N` (struct): Intermediate assembly images with embedded bytes and relative paths

- `final_image` (struct): Complete assembled object image with embedded bytes and relative path

**Quality assurance:** The dataset maintains rigorous quality control through automated validation of mesh integrity, hierarchy consistency, rendering quality, annotation coherence, and multimodal synchronization. We employed 10 professional annotators to manually review critical cases and promote annotation quality. Images are embedded directly in Parquet files for efficient streaming during training.

This Parquet-based packaging enables efficient distributed training with embedded images, supports large-scale multimodal learning, and facilitates reproducible experiments across different object categories. The structured format with embedded visual data helps seamless integration with modern multimodal training pipelines.

## B. Training Details

**Vision tokenization.** For visual generation, we encode each rendered image into VAE (Kingma & Welling, 2013) latents using a pre-trained VAE and keep the VAE frozen during training. The latent features are further patch-embedded to match the Transformer hidden dimension.[1] We also obtain ViT tokens from a ViT encoder (Dosovitskiy et al., 2021) for visual understanding; both ViT and VAE tokens are equipped with 2D positional encoding, and rectified-flow time embeddings are added to the initial hidden states of noised VAE latent tokens.

**Generalized causal attention for interleaving.** Within each sample, tokens are partitioned into multiple consecutive modality splits (text / vision tokens), with the split order following the serialized token stream in Eq. 11. Inside each split, text uses causal attention; vision tokens use bidirectional attention within the split, while respecting the split-level causal ordering (a split cannot attend to future splits). Tokens in one split may attend to all tokens in preceding splits; inside each split, text uses causal attention while vision tokens use bidirectional attention. For each image segment, we maintain three sets of visual tokens: (i) noised VAE tokens used for rectified-flow training, (ii) clean VAE tokens used as conditioning for subsequent segments, and (iii) ViT tokens. Subsequent tokens may attend to the clean VAE and ViT tokens of preceding images, but not to the noised VAE tokens.

**Teacher forcing on interleaved streams.** Given a training instance $(x, \mathcal{T})$ with $\mathcal{T} = \{(z_n, v_n)\}_{n=1}^N$, we serialize it into an interleaved token stream $\mathbf{y} = (y_1, \ldots, y_L)$ as in Eq. 11. We train $\text{ULM}_\theta$ with teacher forcing: at text positions the model predicts the next token under the ground-truth prefix, while at noised-latent positions it predicts the rectified-flow velocity target under the same prefix.

**Token partitions.** Let $\mathcal{V}_{\text{text}}$ denote the unified text vocabulary (including reserved delimiters for multimodal content). We define two disjoint index sets over positions in $\mathbf{y}$:

$$\mathcal{P}_{\text{text}} = \{t \mid y_t \in \mathcal{V}_{\text{text}}\}, \tag{18}$$

$$\mathcal{P}_{\text{mse}} = \{t \mid y_t \text{ corresponds to a noised VAE latent element } \tilde{X}_{n,m}^{(t_n)}\}. \tag{19}$$

Here $\mathcal{P}_{\text{mse}}$ enumerates all positions corresponding to the *noised* VAE latent tokens inside the training-time image-token blocks $V_n = [\tilde{X}_n^{(t_n)}, X_n, U_n]$. We keep the VAE tokenizer components (VAE encoder/decoder) frozen, and fine-tune the Transformer backbone together with the ViT encoder. For notational simplicity, we denote all trainable parameters (Transformer + ViT) by $\theta$.

**Text objective: next-token cross-entropy.** We minimize the standard autoregressive cross-entropy over text-token positions:

$$\mathcal{L}_{\text{CE}}(\theta) = -\sum_{t \in \mathcal{P}_{\text{text}}} \log p_\theta(y_{t+1} \mid y_{\leq t}), \tag{20}$$

where we use the standard one-token shift and exclude the last position from $\mathcal{P}_{\text{text}}$ when forming labels.

**Visual objective: MSE velocity regression.** For each rendered observation $v_n$, we obtain its clean VAE latents $X_n = \text{E}_{\text{vae}}(v_n)$ and sample $t_n \sim \mathcal{U}(0, 1)$ with $\epsilon_n \sim \mathcal{N}(0, I)$, constructing the noised latents along the linear path

$$X_n(t_n) = (1 - t_n)X_n + t_n\epsilon_n. \tag{21}$$

---

[1]Bagel uses a FLUX VAE with downsample ratio 8 and latent channel 16, followed by a $2 \times 2$ patch embedding.

Under this linear path, the target *velocity* is constant:

$$u_n^\star = \epsilon_n - X_n. \tag{22}$$

Let $\hat{u}_t$ be the model prediction at a noised-latent position $t \in \mathcal{P}_{\text{mse}}$ (produced by a regression head on top of $\text{ULM}_\theta$). We minimize the mean-squared error over all noised latent tokens:

$$\mathcal{L}_{\text{MSE}}(\theta) = \sum_{t \in \mathcal{P}_{\text{mse}}} \left\| \hat{u}_t - u_t^\star \right\|_2^2. \tag{23}$$

In practice, each $t \in \mathcal{P}_{\text{mse}}$ corresponds to some latent element $(n, m)$ in $\tilde{X}_n^{(t_n)}$, and $u_t^\star$ denotes the aligned element in $u_n^\star$.

**Final training objective.** We optimize a weighted combination of the text cross-entropy loss and the visual velocity regression loss:

$$\min_\theta \, \mathbb{E}_{(x,\mathcal{T})\sim\mathcal{D}} \Big[ \lambda_{\text{CE}} \, \mathcal{L}_{\text{CE}}(\theta) + \lambda_{\text{MSE}} \, \mathcal{L}_{\text{MSE}}(\theta) \Big], \tag{24}$$

where $\lambda_{\text{CE}}$ and $\lambda_{\text{MSE}}$ are scalar loss weights. We set $\lambda_{\text{CE}} : \lambda_{\text{MSE}} = 1 : 1$ in our experiments to emphasize both text planning quality and visual generation accuracy. In our implementation, $\lambda_{\text{CE}}$ corresponds to `ce_weight` and $\lambda_{\text{MSE}}$ corresponds to `mse_weight`.

**Optimization and decoding.** We train for 2,000 steps with AdamW optimizer ($\beta_1 = 0.9$, $\beta_2 = 0.95$, $\epsilon = 10^{-8}$, weight decay 0), using a cosine learning-rate schedule with 50 warmup steps (peak LR $2 \times 10^{-5}$, min LR $10^{-6}$). We clip gradients to a global norm of $1.0$ and use EMA with decay $0.9999$. Training uses token-budgeted batching with an expected 40K tokens per step and a hard cap of 50K tokens. Each serialized sample is truncated to at most 50K tokens. During packing, if a truncated sample cannot fit into the remaining token budget of the current step, we defer it into an overflow buffer and prioritize sampling from this buffer when the step budget is under 20K tokens. All training is conducted in bfloat16 using FSDP (HYBRID_SHARD; CPU offload enabled).

At inference time, we use classifier-free guidance with text scale 4.0, image scale 2.0, and guidance interval $[0.0, 1.0]$. To support CFG, we apply the same modality-dropout to the conditioning prefix during training, dropping text, VAE, and ViT conditioning tokens with probabilities 0.1, 0.3, and 0.3, respectively, to construct unconditional prefixes. We compute conditional and unconditional predictions and apply CFG as $\hat{u} = u_{\text{uncond}} + s_{\text{img}}(u_{\text{cond}} - u_{\text{uncond}})$ for RF velocity, and similarly for text logits with scale $s_{\text{text}}$.

We alternate between (i) sampling text plans autoregressively with temperature 0.3 (Eq. 12) and (ii) sampling VAE latents with rectified flow using 50 timesteps and timestep shift 1.0 (Eq. 13), followed by decoding through the frozen VAE decoder (Eq. 14). After obtaining $\hat{v}_n$, we set $X_n = \hat{X}_n$ and compute $U_n = \text{E}_{\text{vit}}(\hat{v}_n)$ as deterministic conditioning tokens for subsequent steps.

## C. Inference Termination Mechanism

The Shape-of-Thought (SoT) framework employs a carefully designed termination mechanism to determine when the progressive assembly process should conclude. This mechanism ensures that the model generates an appropriate number of assembly steps and produces a coherent final output without over-generation or premature termination.

Our termination mechanism consists of two complementary signals:

**Structural Termination Marker (Training).** During training, we use a structured termination marker to teach the model when assembly is complete. Each training example contains a `Final Assembly` field formatted as `<assembly>Final Assembly: FINISH</assembly>`. This marker serves as a clear signal that the progressive assembly process has reached its conclusion. The model learns to predict this token sequence at the appropriate moment, effectively learning when enough structural components have been assembled to satisfy the goal prompt.

The training data constructs interleaved sequences where each reasoning step text is wrapped in `<thought>` and `</thought>` tokens, each image is bounded by `<image_start>` and `<image_end>` tokens, and the final assembly confirmation is wrapped in `<assembly>` and `</assembly>` tokens.

*Table 4.* Training hyperparameters (aligned with the main-text Optimization paragraph).

| Item | Value |
|---|---|
| Backbone | Bagel-7B (MoT; `Qwen2MoTDecoderLayer`) |
| Training steps | 2,000 |
| Optimizer | AdamW ($\beta_1$=0.9, $\beta_2$=0.95, $\epsilon$=$10^{-8}$) |
| Weight decay | 0 |
| LR schedule | cosine |
| Warmup steps | 50 |
| Peak LR / min LR | $2 \times 10^{-5}$ / $10^{-6}$ |
| Grad clip | 1.0 (global norm) |
| EMA decay | 0.9999 |
| Loss weights (CE / MSE) | 1 / 1 |
| Token budget (expected / max) | 40K / 50K tokens per step |
| Max tokens / sample | 50K (truncate; overflow-buffer packing for budget overflow) |
| Max latent size | 64 |
| Precision | bfloat16 |
| Sharding | FSDP, HYBRID_SHARD (32 shards; CPU offload enabled) |
| Conditioning dropout (text / VAE / ViT) | 0.1 / 0.3 / 0.3 |
| Frozen modules | VAE (encoder/decoder/tokenizer) |
| Trainable modules | Transformer backbone + ViT encoder |
| Seeds (data / global) | 42 / 4,396 |
| Infrastructure | 32× NVIDIA H100 80GB; Python 3.10.19; Linux (glibc 2.35) |

**Token-Level Termination (Inference).** At inference time, we employ a token-level termination detection mechanism implemented in the `generate_text` function. The model uses the end-of-sequence token (`<|im_end|>`) as the primary termination signal. When this token is generated, the termination mechanism performs an additional lookahead step: if the next token is `<|vision_start|>`, the generation continues to produce the corresponding image; otherwise, the generation loop terminates. A maximum token limit (`max_length`) provides a hard cap to prevent infinite generation loops.

This lookahead mechanism is essential because the end-of-sequence token may appear in two distinct contexts: (1) **True Termination**: the model has completed assembly and signals the end of generation; (2) **Image Block Transition**: the model signals the end of the current text block and expects to generate the next image.

The termination mechanism enables SoT to automatically determine the appropriate number of assembly steps for each object, adapting to the inherent complexity of the target structure without requiring explicit step-count specification from the user.

# D. Experimental Details

## D.1. Hyperparameters and infrastructure.

We fine-tune Bagel-7B end-to-end with AdamW ($\beta_1$=0.9, $\beta_2$=0.95, $\epsilon$=$10^{-8}$) for 2,000 optimization steps. We use a cosine learning-rate schedule with 50 warmup steps (peak LR $2 \times 10^{-5}$, min LR $10^{-6}$), global gradient clipping at 1.0, and exponential moving average (EMA) with decay 0.9999. Training uses bfloat16 with FSDP under a HYBRID_SHARD strategy (32 shards) and CPU offloading enabled. We adopt token-budgeted batching with an expected 40K tokens per step and a hard cap of 50K tokens, truncating each serialized sample to at most 50K tokens (and skipping over-limit samples with an overflow buffer). We freeze the VAE while fine-tuning the Transformer and the ViT modules; dropout probabilities are 0.1 for text conditioning and 0.3 for VAE/ViT conditioning. All experiments are run with fixed data/global seeds (42 / 4,396) on 32× NVIDIA H100 (80GB) GPUs.

## D.2. Training Metrics Visualization

Figure 7 presents the evolution of key training metrics over the course of 2,000 optimization steps. The visualization provides insights into the stability and convergence behavior of our model:

- **Data Throughput and Tokens:** Subfigure (a) illustrates the *Total Samples per Step*, which, after an initial startup

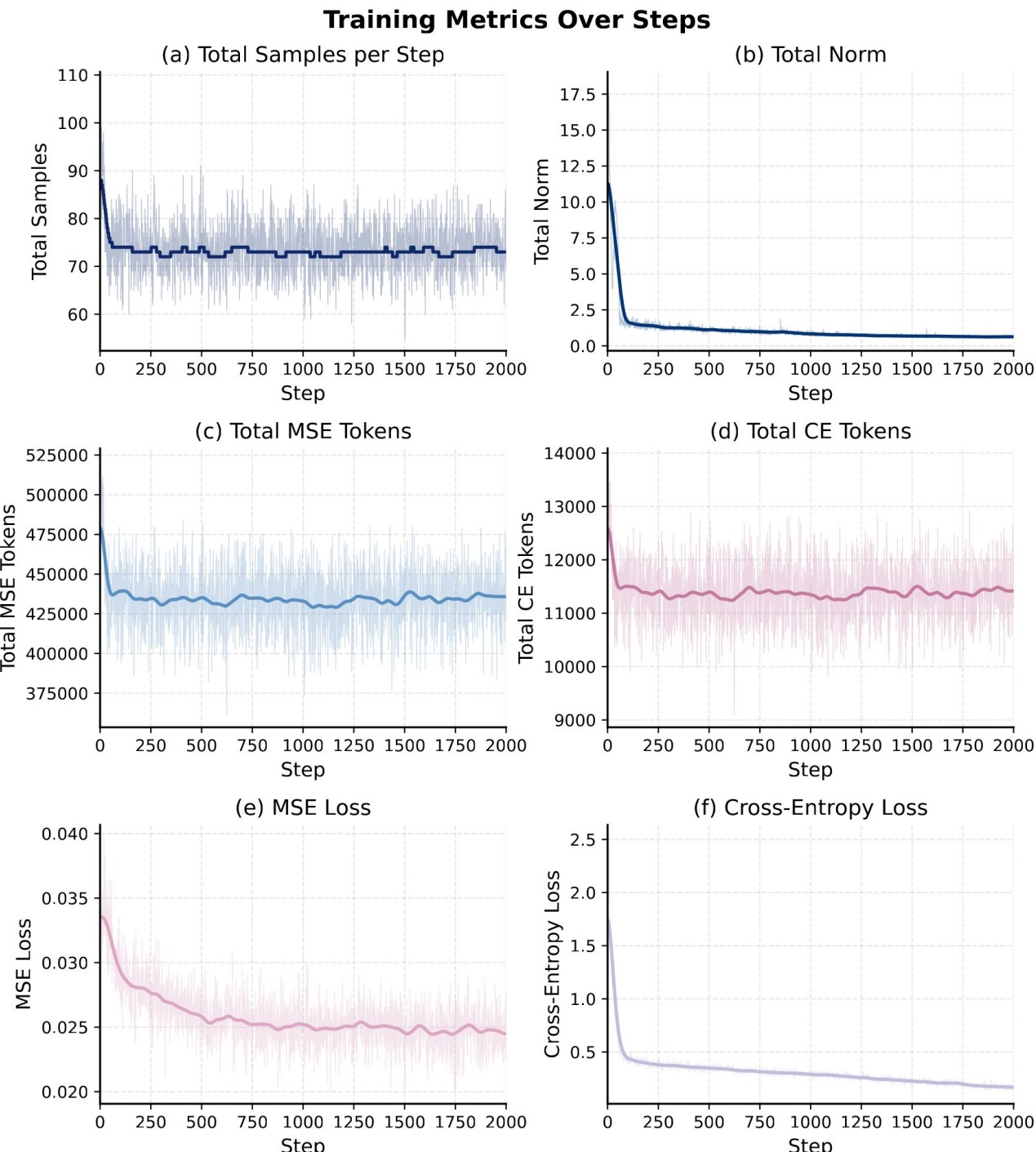

*Figure 7.* **Training Metrics Over Steps.** Complete training visualization including: (a) Total samples per step, (b) Total norm, (c-d) MSE and CE token counts, (e-f) Loss curves.

phase, stabilizes at approximately 73 samples per step. Similarly, subfigures (c) and (d) show the *Total MSE Tokens* and *Total CE Tokens* respectively. These metrics remain consistent throughout the training process (averaging roughly 435,000 MSE tokens and 11,400 CE tokens per step), indicating a stable data pipeline and consistent batch composition.

- **Optimization Stability:** Subfigure (b) displays the *Total Norm* of the gradients. We observe a sharp initial decrease from a peak of over 12.5 to below 2.5 within the first 100 steps, followed by a gradual, smooth decline. This behavior suggests that the optimization process is well-conditioned and free from significant gradient explosions, leading to stable parameter updates.

- **Loss Convergence:** The primary objectives, *MSE Loss* (e) and *Cross-Entropy (CE) Loss* (f), both demonstrate clear downward trajectories. The MSE loss steadily decreases from 0.035 and begins to plateau around 0.025. The CE loss exhibits a more rapid initial descent from 2.5, continuing a sustained decline towards 0.2 by the end of 2,000 steps. The lack of significant spikes in either loss curve confirms that the learning rate schedule and optimization strategy are effective for joint regression and classification tasks.

Overall, these metrics confirm a healthy training regime with consistent token throughput and steady convergence across multiple objective functions.

## E. T2S-CompBench: Implementation Details

### E.1. Pilot Human Audit for Protocol Selection

To choose reliable evaluators in our rendered domain, we conducted a pilot human audit on fine-grained component recognition. We selected 24 component sub-classes and sampled 10 rendered images per class. For each image, we derive per-component text prompts by parsing its `prompt.json` (one query per component), and evaluate three evaluators: (i) SAM 3 (Carion et al., 2025) concept segmentation, (ii) Grounded-SAM (Ren et al., 2024), and (iii) GPT-4o (Hurst et al., 2024) judging.

For SAM-based methods, a component is considered *recognized* if the method returns at least one valid mask after thresholding; for GPT-4o, recognition is positive if the predicted count is $> 0$. We compute precision/recall over component queries, where false positives correspond to recognizing the wrong component (or returning an invalid region), and false negatives correspond to failing to recognize an existing component. The audit shows that VLM judging is more reliable for fine-grained component recognition/counting, whereas SAM 3 remains accurate for extracting stepwise *union foreground* masks. This motivates our hybrid protocol in T2S-CompBench (VLM for semantics; masks for geometry).

*Table 5.* Pilot audit accuracy (%) on fine-grained component recognition (24 sub-classes, 10 images each).

| Method | Precision | Recall | F1-Score |
|---|---|---|---|
| SAM 3 (Text Prompt) | 100.00 | 39.05 | 56.17 |
| Grounded-SAM | 100.00 | 41.76 | 58.92 |
| **GPT-4o (Ours)** | **99.51** | **94.23** | **96.80** |

### E.2. GPT-4o Judge: Binary Forcing and Confidence

We use GPT-4o via API with deterministic decoding (temperature $= 0$) and enforce closed-form outputs: `Yes/No` for semantic questions and `Attached/Detached` for connectivity. We request answer-token log-scores and compute two-way softmax confidence. For a binary decision with answer options $\{u, v\}$ and corresponding token log-scores $(\ell_u, \ell_v)$, we define

$$\text{Conf}_u = \frac{\exp(\ell_u)}{\exp(\ell_u) + \exp(\ell_v)}. \tag{25}$$

To improve parse reliability for counting, we optionally use structured outputs (JSON schema) and extract the integer field `count`.

### E.3. Instruction Parsing and Metric Definitions (M1–M7)

We parse the goal instruction $x$ into: required component categories $\mathcal{C}$ with required counts $\{N_c^{\mathrm{req}}\}$, part-bound attribute items $\mathcal{A}$, required connectivity pairs $\mathcal{E}$, and relation triplets $\mathcal{R}$. Unless otherwise specified, semantic/relational metrics are computed by forced-choice GPT-4o judging using Eq. (25) under criteria-based prompt templates (Sec. E.6).

**M1: CN (Component Integrity & Numeracy).** To penalize missing parts and over/under-counting under compositional goals, we score each required category by a scale-invariant count accuracy and then average. Concretely, for each $c \in \mathcal{C}$ we query GPT-4o to estimate an integer count $N_c^{\mathrm{pred}}$ and gate by presence $\mathrm{Recall}_c = \mathbb{I}[N_c^{\mathrm{pred}} > 0]$, so that a missing required category contributes zero. If present, we assign a per-category score that decays linearly with *relative* count error and is clipped to $[0, 1]$:

$$\mathrm{CN} = \frac{1}{|\mathcal{C}|} \sum_{c \in \mathcal{C}} [\mathrm{Recall}_c] \cdot \max\left(0,\ 1 - \frac{|N_c^{\mathrm{pred}} - N_c^{\mathrm{req}}|}{\max(1, N_c^{\mathrm{req}})}\right). \tag{26}$$

The normalization by $\max(1, N_c^{\mathrm{req}})$ makes the penalty comparable across categories with different required counts and avoids division by zero, and the outer $\max(0, \cdot)$ prevents negative scores when the count mismatch is large.

**M2: SF (Shape & Category Fidelity).** To detect failures where parts appear plausible but the overall body shape/category is wrong, we ask a forced `Yes`/`No` question $q_s$ derived from $x$ and score

$$\mathrm{SF} = \mathrm{Conf}_{\texttt{Yes}}(q_s). \tag{27}$$

**M3: AF (Attribute Fidelity).** To check part-level attribute binding (e.g., "square handle", "cylindrical cap"), we ask a forced `Yes`/`No` query $q_a$ for each attribute item $a \in \mathcal{A}$ and average:

$$\mathrm{AF} = \frac{1}{|\mathcal{A}|} \sum_{a \in \mathcal{A}} \mathrm{Conf}_{\texttt{Yes}}(q_a), \tag{28}$$

where each $a$ specifies a target category $c(a)$ and an attribute; if the target category is missing (i.e., $N_{c(a)}^{\mathrm{pred}} = 0$ in CN), we set that term to $0$.

**M4: CP (Connectivity Plausibility).** To test whether the object reflects *assembly* rather than a collage of floating parts, we test whether each required connectivity pair $(A, B) \in \mathcal{E}$ appears attached by a forced `Attached`/`Detached` query and compute

$$\mathrm{CP} = \frac{1}{|\mathcal{E}|} \sum_{(A,B) \in \mathcal{E}} \mathrm{Conf}_{\texttt{Attached}}(A, B). \tag{29}$$

**M5: VT (Visual Topology).** To evaluate spatial relations that require 3D-aware reasoning even under a single canonical view (e.g., above/below/inside/left-of), we ask a forced `Yes`/`No` relation query $q_r$ for each triplet $r \in \mathcal{R}$ and average:

$$\mathrm{VT} = \frac{1}{|\mathcal{R}|} \sum_{r \in \mathcal{R}} \mathrm{Conf}_{\texttt{Yes}}(q_r). \tag{30}$$

Each $r$ encodes a relation (subject, predicate, object) parsed from $x$.

**M6: TS (Trace Stability).** To quantify unintended global re-drawing or geometric drift across steps, we measure how much of the previously visible structure is preserved after each incremental update. Let $M^{(n)}$ be the SAM 3 *union* foreground mask extracted from step image $v_n$ (front view), which approximates the visible silhouette of the cumulative assembly structure at step $n$. We compute stepwise retention of the previous silhouette and average across steps:

$$\mathrm{TS} = \frac{1}{N-1} \sum_{n=2}^{N} \frac{|M^{(n)} \cap M^{(n-1)}|}{\max(1,\ |M^{(n-1)}|)}, \tag{31}$$

where $|\cdot|$ denotes pixel area.

**M7: RA (Stepwise Rationale Alignment).** To evaluate whether each rationale $z_n$ explains the main visual change from $v_{n-1}$ to $v_n$, we construct a forced `Yes`/`No` question $q_n$ from `step_n.json` (the intended newly added leaf part and its attachment), query GPT-4o on the image pair $(v_{n-1}, v_n)$, and average:

$$\text{RA} = \frac{1}{N-1} \sum_{n=2}^{N} \text{Conf}_{\texttt{Yes}}(q_n). \tag{32}$$

### E.4. Multi-view Aggregation Protocol (T2S-CompBench-MV)

**Views.** When multi-view predictions are available, we evaluate four fixed views $\mathcal{V} = \{\texttt{front}, \texttt{left}, \texttt{right}, \texttt{back}\}$. For each metric $m$ and view $v \in \mathcal{V}$, we compute a view-wise score $m^{(v)}$ using the same procedure as the single-view setting, but feeding the corresponding view image(s).

**Aggregation.** We adopt a view-aggregation rule that distinguishes *visibility-sensitive* metrics from *global-consistency* metrics:

$$
\begin{aligned}
m^{\text{MV}} &= \max_{v \in \mathcal{V}} m^{(v)}, && m \in \{\text{CN}, \text{AF}, \text{CP}\}, \\
m^{\text{MV}} &= \frac{1}{|\mathcal{V}|} \sum_{v \in \mathcal{V}} m^{(v)}, && m \in \{\text{SF}, \text{VT}, \text{TS}, \text{RA}\}.
\end{aligned}
\tag{33}
$$

**Why max for CN/AF/CP (presence-sensitive).** These metrics depend on whether a *specific component* (and its local property/attachment) is *visible* in a 2D projection. Occlusion and foreshortening can induce false negatives in a single view, while multi-view observations provide redundancy and mitigate occlusion failures.(Xie et al., 2025; Asim et al., 2025) **CN (counting / recall).** Consider a sword: in a side view the blade may collapse to a thin line, causing the judge to output $N_{\text{blade}}^{(\texttt{side})} = 0$ and thus $\text{CN}^{(\texttt{side})} = 0$. Max-pooling implements an "exist-if-visible-anywhere" rule: $\max(\text{CN}^{(\texttt{front})}, \text{CN}^{(\texttt{side})})$ restores the correct presence and avoids occlusion-induced false negatives. **AF (part-bound attributes).** Attributes such as "square handle" are only verifiable when the target part is visible with sufficient shape evidence; max-pooling prevents penalizing a correct attribute simply because the part is self-occluded in one view. **CP (connectivity).** Whether two parts appear attached is also visibility-dependent (the contact region may be hidden in a given view); max-pooling avoids spurious "Detached" decisions caused by view-specific occlusion.

**Why mean for SF/VT/TS/RA (global consistency).** These metrics are intended to capture *view-consistent 3D plausibility* rather than view-specific visibility. Recent multi-view generation benchmarks and metrics emphasize that 3D consistency across views is a central requirement for geometry-aware generation.(Xie et al., 2025; Asim et al., 2025; He et al., 2023) Mean aggregation penalizes "view-inconsistent" artifacts that can look correct from one view but fail from others. **SF (global body shape).** A model should not satisfy the shape scaffold only from a single view; averaging encourages a body geometry that remains plausible under viewpoint change. **VT (spatial relations).** Relations like `on-top-of` can be ambiguous in a single 2D projection (e.g., due to perspective overlap). Averaging over multiple views approximates multi-angle sampling of the underlying 3D relation: to achieve a high mean score, the relation must be consistently supported across viewpoints, reducing projection ambiguity. **TS (trace stability).** Mean aggregation is crucial to penalize "billboard / paper-thin" failures: an object may appear stable from `front` (high $\text{TS}^{(\texttt{front})}$) but collapse or deform under `side` (low $\text{TS}^{(\texttt{side})}$). The mean yields a medium score, exposing view inconsistency, which is widely recognized as a key failure mode in multi-view/3D generation.(Xie et al., 2025) **RA (stepwise rationale alignment).** If the step rationale describes a 3D-consistent structural constraints addition, the *main change* should be coherent across views over the trace. Mean aggregation discourages view-specific hallucinated edits (e.g., a part appears only in one view) and rewards stepwise changes that remain explainable under viewpoint change.

### E.5. M1 (CN): Per-category Counting Prompts

For each required category $c$, we query GPT-4o *separately* to avoid cross-category interference. We constrain the output to an integer (or JSON $\{\texttt{count}: k\}$). If the response is invalid, we re-ask with an explicit format reminder and take the first valid output. We use the resulting $N_c^{\text{pred}}$ to compute CN in Eq. (26).

### E.6. M2–M5: Rigorous Criteria-Based Prompt Templates

Inspired by the rigorous protocol of T2I-CompBench++ (Huang et al., 2025), we adopt a **criteria-based prompting strategy**. Instead of open-ended queries, we provide the evaluator with explicit definitions of positive and negative classes to standardize the decision boundary. All evaluations share a common system prompt to establish the auditing persona.

We provide the detailed prompt templates for the System Prompt and Metrics M2–M5 in Appendix M.3, including:

- **M2 (SF):** Topology audit for global shape matching

- **M3 (AF):** Local property verification for part-bound attributes

- **M4 (CP):** Physical assembly audit for connectivity validation

- **M5 (VT):** 3D spatial reasoning for relation verification

### E.7. M6 (TS): Stepwise Union Foreground Masks with SAM 3

We extract a union foreground mask $M^{(n)}$ from each step image $v_n$ using SAM 3. Since fine-grained part classification is unreliable in our domain, we do not require part-level masks for TS. When multiple masks are produced, we take the union of masks with confidence above a fixed threshold and (optionally) keep the largest connected component as the main object mask. We compute TS as the mean retention ratio in Eq. (31), which measures how much of the previous mask is preserved by the current mask.

### E.8. M7 (RA): Stepwise Faithfulness Questions

Let $z_n$ be the rationale in `step_n.json`. We query GPT-4o with the image pair $(v_{n-1}, v_n)$ and ask whether the *main* visual change is adding the component/action described by $z_n$. We force a `Yes/No` answer and compute $\text{Conf}_{\text{Yes}}(q_n)$ for RA in Eq. (32).

## F. Budget-Controlled 2D Baselines

To test whether SoT's gains are merely caused by more intermediate steps or more image decodes, we compare against two budget-controlled 2D baselines. *CoT-K* generates $K$ textual reasoning steps followed by one final image, while *Refine-K* performs $K$ iterative image re-decodes without interleaved text-image traces. The structure average is computed over CN/AF/CP/VT in this controlled run. Even when Refine-K closely matches SoT in image decodes and latency, SoT remains substantially stronger, especially on connectivity and topology.

*Table 6.* **Budget-controlled 2D comparison.** CoT-K and Refine-K control for extra reasoning steps and image decoding budget.

| Method | Text Steps | Image Decodes | Latency (s) | CN | AF | CP | VT | Avg. |
|---|---|---|---|---|---|---|---|---|
| Bagel-7B (Direct) | 1.00 | 1.00 | 52.35 | 64.81 | 58.92 | 62.63 | 65.77 | 63.03 |
| Bagel-7B-CoT (orig.) | 2.45 | 2.45 | 102.96 | 75.54 | 71.93 | 68.57 | 71.11 | 71.79 |
| Bagel-7B-CoT-K | 5.64 | 1.00 | 79.82 | 76.19 | 72.56 | 68.98 | 72.23 | 72.49 |
| Bagel-7B-Refine-K | 5.76 | 5.76 | 248.97 | 78.41 | 73.85 | 69.62 | 72.79 | 73.67 |
| **Bagel-7B-SoT** | **5.83** | **5.83** | **259.13** | **88.17** | **81.74** | **85.96** | **84.48** | **85.09** |

## G. T2S-CompBench-MV Extension

We additionally evaluate rendered multi-view outputs using the aggregation protocol in Appendix E.4. The zero-shot SoT row evaluates the original single-view SoT model under the multi-view evaluation protocol, without multi-view fine-tuning. The MV-finetuned row illustrates how the trace formulation can incorporate additional view supervision.

Qualitative multi-view examples are provided with the additional qualitative results in Appendix L. The corresponding failure mode is discussed separately in Appendix J.

*Table 7.* **T2S-CompBench-MV.** Multi-view rendered evaluation over front/left/right/back views. RA/TS require trace outputs and are not available for final-asset-only baselines.

| Method | CN | SF | AF | CP | VT | RA | TS | Latency (s) |
|---|---|---|---|---|---|---|---|---|
| Bagel-7B (Zero-shot) | 55.12 | 52.34 | 48.21 | 50.11 | 45.67 | – | – | 52.14 |
| Bagel-7B-CoT (Zero-shot) | 62.45 | 54.12 | 60.33 | 55.42 | 51.28 | 30.12 | 18.55 | 103.21 |
| Bagel-7B-SoT (Zero-shot) | 78.41 | 68.22 | 73.15 | 76.88 | 69.45 | 62.14 | 70.33 | 258.46 |
| Shap-E | 41.11 | 67.91 | 24.27 | 20.08 | 21.05 | – | – | 10.07 |
| LGM | 67.68 | 72.43 | 54.38 | 71.24 | 68.72 | – | – | 6.57 |
| L3GO | 75.22 | 58.15 | 67.41 | 59.13 | 65.32 | – | – | 927.84 |
| Meshy 6 | 81.76 | 87.86 | 74.22 | 84.55 | 70.61 | – | – | 75.36 |
| Bagel-7B-SoT (MV Finetuned) | **87.42** | 76.14 | **80.45** | **85.21** | **77.24** | **71.65** | **83.72** | 259.02 |

# H. Human Evaluation

We conducted human evaluation via Wen JuanXing (WJX). For each test prompt, annotators were asked to rate the generated results on three aspects: *Visual Quality*, *Structural Correctness*, and *Progressive Assembly Logic*. Figures 8, 9, 10, 11, 12, and 13 show the interface used for human evaluation. We randomly sample 50 test prompts stratified by object category. For each prompt, we present outputs from different methods in a blind and randomized order (method identity hidden), and collect 3 independent ratings per method by randomly drawing annotators from the pool. In total, we gather 1050 prompt–output rating instances for human evaluation. Each prompt–output pair is rated by 3 annotators with a score from 1 to 5 for each aspect. Each aspect is rated independently on a 5-point scale: (1) completely not satisfied; (2) partially satisfied; (3) roughly satisfied ($\sim$50%); (4) largely satisfied; (5) fully satisfied. We recruited 50 paid annotators. Annotators are paid 0.5 RMB per image evaluated (per final output), and we spend 800 RMB in total on participant compensation.

Text prompt: **Construct a compact round planter with a sturdy cylindrical base filled with soil, topped by tall, slender leaves extending upward in a clustered arrangement.**

Generate image:

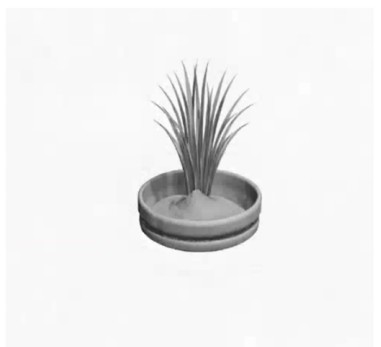

Rate the matching degree of **Visual Quality**, **Structural Correctness**, and **Progressive Assembly Logic** between the Image and Text Prompt

◯ **5 - Fully satisfied :** means fully aligned with the prompt/criterion

◯ **4 - Largely satisfed :** means largely aligned with minor issues

◯ **3 - Roughly satisfed (~50%) :** means partially aligned

◯ **2 - Partially satisfied :** means mostly misaligned

◯ **1 - Completely not satisfied :** means almost irrelevant

*Figure 8.* WJX interface for the image-text alignment evaluation on SoT (Vase).

Text prompt: **Construct a sleek, S-shaped chair with a continuous curved base forming a seamless transition into the seat and back, featuring a single smooth surface design without visible joints or separations.**

Generate image:

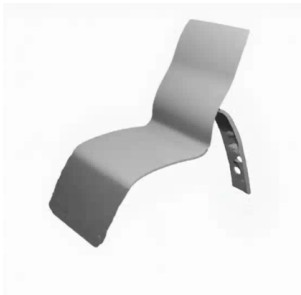

Rate the matching degree of *Visual Quality*, *Structural Correctness*, and *Progressive Assembly Logic* between the Image and Text Prompt

○ **5 - Fully satisfied :** means fully aligned with the prompt/criterion

○ **4 - Largely satisfed :** means largely aligned with minor issues

○ **3 - Roughly satisfed (~50%) :** means partially aligned

○ **2 - Partially satisfied :** means mostly misaligned

○ **1 - Completely not satisfied :** means almost irrelevant

*Figure 9.* WJX interface for the image-text alignment evaluation on SoT (Chair).

Text prompt: **Construct a modern faucet with an angular, L-shaped spout, mounted on a rectangular base, featuring two hexagonal switches on either side of the spout for controlling water flow.**

Generate image:

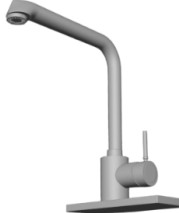

Rate the matching degree of **Visual Quality**, **Structural Correctness**, and **Progressive Assembly Logic** between the Image and Text Prompt

○ **5 - Fully satisfied :** means fully aligned with the prompt/criterion

○ **4 - Highly satisfied :** means largely aligned with minor issues

○ **3 - Moderately satisfied :** means partially aligned

○ **2 - Slightly satisfied :** means mostly misaligned

○ **1 - Not satisfied :** means almost irrelevant

*Figure 10.* WJX interface for the image-text alignment evaluation on Meshy6 (Faucet).

Text prompt: **Construct a modern, minimalist chair featuring a continuous S-shaped form with a single surface seat seamlessly connecting to a vertically aligned backrest, both supported by a flat, rectangular base.**

Generate image:

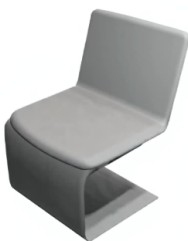

Rate the matching degree of *Visual Quality*, *Structural Correctness*, and *Progressive Assembly Logic* between the Image and Text Prompt

○ **5 - Fully satisfied :** means fully aligned with the prompt/criterion

○ **4 - Largely satisfed :** means largely aligned with minor issues

○ **3 - Roughly satisfed (~50%) :** means partially aligned

○ **2 - Partially satisfied :** means mostly misaligned

○ **1 - Completely not satisfied :** means almost irrelevant

*Figure 11.* WJX interface for the image-text alignment evaluation on SoT (Chair).

Text prompt: **Construct a rectangular ping pong table featuring a smooth, flat tabletop surface with a centered net divider, supported by four tapered legs connected by bar stretchers for stability.**

Generate image:

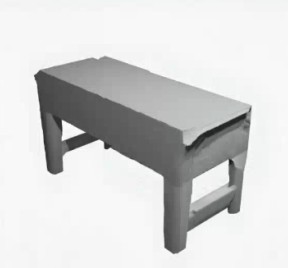

Rate the matching degree of *Visual Quality*, *Structural Correctness*, and *Progressive Assembly Logic* between the Image and Text Prompt

○ **5 - Fully satisfied :** means fully aligned with the prompt/criterion

○ **4 - Largely satisfed :** means largely aligned with minor issues

○ **3 - Roughly satisfed (~50%) :** means partially aligned

○ **2 - Partially satisfied :** means mostly misaligned

○ **1 - Completely not satisfied :** means almost irrelevant

*Figure 12.* WJX interface for the image-text alignment evaluation on SoT (Table).

Text prompt: **Construct a spherical lamp cover featuring evenly spaced vertical slats curving outward from a narrow top rim to a wider open base, creating a cage-like structure.**

Generate image:

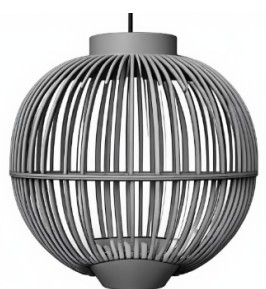

Rate the matching degree of **Visual Quality**, **Structural Correctness**, and **Progressive Assembly Logic** between the Image and Text Prompt

○ **5 - Fully satisfied :** means fully aligned with the prompt/criterion

○ **4 - Largely satisfed :** means largely aligned with minor issues

○ **3 - Roughly satisfed (~50%) :** means partially aligned

○ **2 - Partially satisfied :** means mostly misaligned

○ **1 - Completely not satisfied :** means almost irrelevant

*Figure 13.* WJX interface for the image-text alignment evaluation on Bagel (Lamp).

**Aggregation and uncertainty.** We first average the 3 ratings per prompt to obtain a prompt-level score, and then report the mean across prompts. We additionally report **mean $\pm$ 95% CI** computed via **prompt-level bootstrap**: we resample prompts with replacement, recompute the mean, and take the 2.5/97.5 percentiles as the 95% CI.

**Inter-annotator agreement.** We compute **Fleiss' $\kappa$** *separately for each aspect* by treating the 5-point ratings as categorical labels with $n = 3$ raters per prompt.

## I. Judge Diversity and Consistency

**Motivation.** VLM-as-a-judge may introduce model-specific biases. We therefore validate the robustness of T2S-CompBench by re-auditing a stratified subset with an alternative open-weight VLM judge and quantifying cross-judge consistency.

**Judges.** Our primary judge is GPT-4o (Sec. E.2). We additionally use **Qwen2-VL-7B-Instruct** (Wang et al., 2024) as an alternative multimodal judge.

**Subset and protocol.** We sample $N_{\text{sub}}$=200 test prompts stratified by object category. For each prompt, we re-run the judge-based metrics **CN/SF/AF/CP/VT/RA** on the corresponding rendered outputs (using the identical prompt templates and forced-choice formats). Metric **TS** is mask-based (SAM 3) and thus judge-independent.

**Closed-form outputs and confidence.** For binary questions (SF/AF/VT/RA: `Yes`/`No`; CP: `Attached`/`Detached`), we enforce closed-form outputs with deterministic decoding. As Qwen2-VL local inference does not reliably expose answer-token log-scores, we approximate confidence by **self-consistency voting**: repeat each query $R$=5 times (temperature $= 0.2$, top-$p = 0.9$) and define $\widehat{\text{Conf}}_{\text{Yes}} = \frac{1}{R}\sum_{r=1}^{R}\mathbb{I}[\text{Yes}]$ (and similarly for `Attached`). We use the same confidence definition for GPT-4o by mapping its forced-choice output to $\{0, 1\}$ when computing agreement; score-level correlations use confidence-like scores in $[0, 1]$.

*Table 8.* **Qwen re-audit setup for judge consistency.**

| Item | Value |
|---|---|
| Primary judge | GPT-4o |
| Alternative judge | Qwen2-VL-7B-Instruct |
| Subset sampling | $N_{sub} = 200$ prompts, stratified by category |
| Judged metrics | CN/SF/AF/CP/VT/RA (TS is mask-based) |
| Binary forcing | Yes/No; Attached/Detached |
| Confidence (Qwen) | self-consistency voting ($R = 5$) |
| Uncertainty | 95% CI via prompt-level bootstrap |

*Table 9.* **Cross-judge consistency on the stratified subset.**

| Metric | Spearman $\rho$ ↑ | Kendall $\tau$ ↑ | Agree (%)↑ | Cohen $\kappa$ ↑ | 95% CI |
|---|---|---|---|---|---|
| CN | 0.65 | 0.51 | 74.5 | 0.48 | ±0.09 |
| SF | 0.84 | 0.68 | 88.0 | 0.69 | ±0.06 |
| AF | 0.76 | 0.59 | 81.2 | 0.58 | ±0.07 |
| CP | 0.72 | 0.56 | 77.4 | 0.53 | ±0.08 |
| VT | 0.69 | 0.53 | 75.8 | 0.50 | ±0.08 |
| RA | 0.79 | 0.62 | 83.6 | 0.61 | ±0.06 |
| **Method ranking** | Rank $\tau$: 0.86 | | Top-1 Match: 100% | | – |

**Consistency metrics.** Let $s_i^{(a)}$ be the per-sample score from judge $a$ for a given metric. We report: (i) **score correlation** (Spearman $\rho$, Kendall $\tau$) between $\{s_i^{(\text{GPT-4o})}\}$ and $\{s_i^{(\text{Qwen})}\}$; (ii) **binary decision agreement** by thresholding at 0.5 to get $y_i \in \{0, 1\}$, then reporting raw agreement and Cohen's $\kappa$; (iii) **method-level ranking stability** by ranking methods using subset-level mean scores and reporting Kendall $\tau$ between rankings as well as the top-1 match rate. All statistics are reported with **95% CIs** via bootstrap resampling over prompts (2.5/97.5 percentiles).

**Qualitative disagreement audit.** To diagnose systematic judge differences, we further inspect the top-10 prompts with the largest absolute score gaps $|s_i^{(\text{GPT-4o})} - s_i^{(\text{Qwen})}|$ per metric and categorize failure modes (e.g., occlusion sensitivity for CN/AF/CP, ambiguous projection for VT, and small-step edits for RA). These cases are used as diagnostic checks for prompt sensitivity and evaluator disagreement.

# J. Limitations

**Failure cases.** Figure 14 shows representative examples where SoT produces globally coherent assemblies but still misses small or thin components. These errors are most prominent when the missing parts are partially or fully occluded in the canonical front view, or when they project to sub-pixel / low-contrast regions. This suggests that single-view rendered supervision can under-specify occluded geometry: the model may satisfy the observed 2D evidence while failing to recover hidden details, leading to residual part omissions despite correct high-level topology.

Multi-view diagnostics expose the same limitation more directly. Figure 15 shows a case where the canonical front view appears plausible, but side/back views reveal incorrect geometry or view-specific structural errors. This illustrates how single-view rendered supervision can under-specify hidden structure and motivates stronger view-consistent supervision.

**Viewpoint and Modality Constraints.** The main SoT training setting is centered on a canonical front view, a choice dictated by the prohibitive computational cost of dense visual traces (which already saturate 32×H100 GPUs in the single-view setting). Naively scaling dense multi-view supervision to every trace step would multiply visual token consumption and training time, rendering it infeasible under current constraints. SoT-26K includes auxiliary left/right/back renderings, our codebase supports multi-stream data loading, and T2S-CompBench-MV provides a rendered multi-view aggregation protocol. Appendix G reports an MV-finetuned variant, and compute-efficient strategies such as stochastic view sampling or latent distillation are natural directions for robust multi-view training.

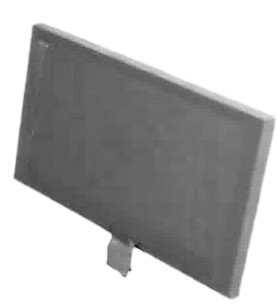

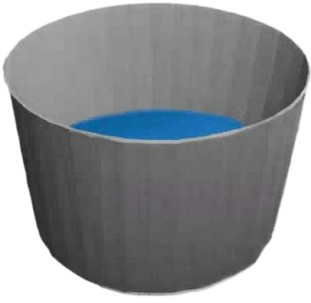

*(a)* Case A: The model did not generate the base of the display.

*(b)* Case B: The model generated an incorrect shape due to the occlusion.

*Figure 14.* **Representative SoT failure cases.** Single-view rendered supervision may miss occluded or low-contrast parts, causing omitted details or incorrect shapes despite a plausible global object.

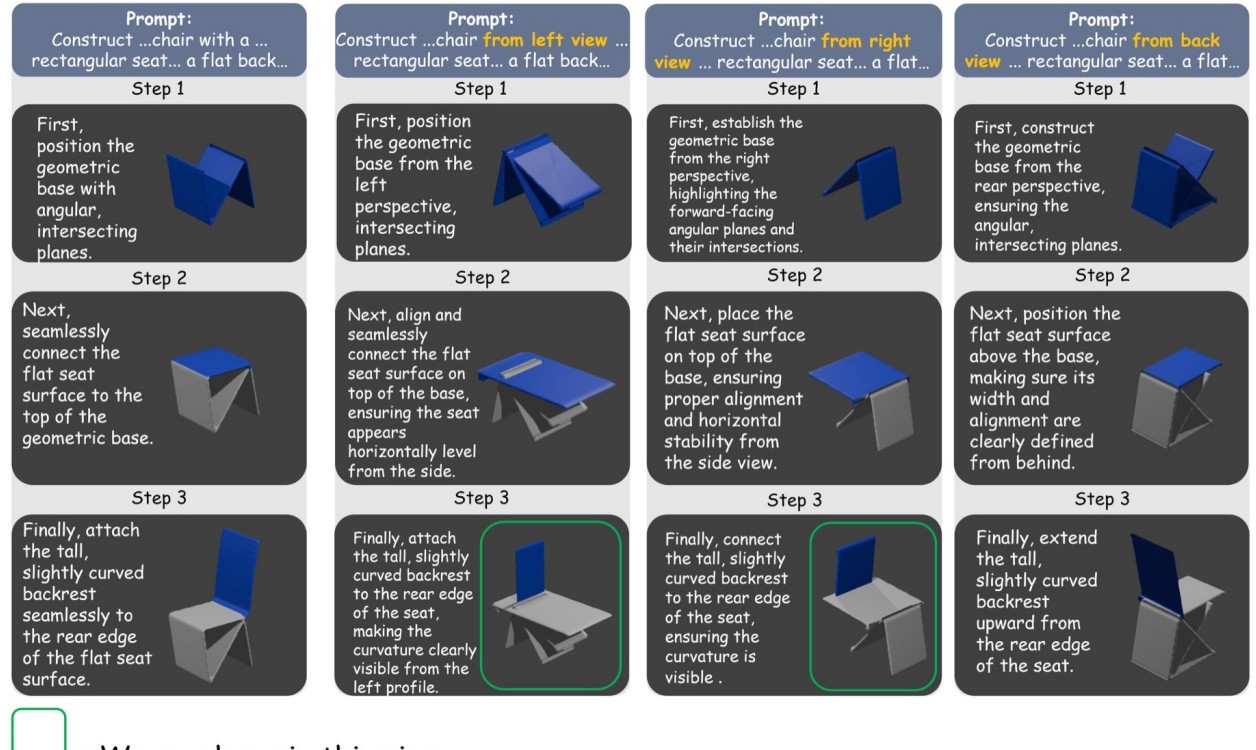

*Figure 15.* **Multi-view failure case.** Some outputs look plausible from the canonical front view but reveal incorrect geometry or view-specific structural errors from side or back views, motivating stronger multi-view supervision.

**Compute Cost.** Our current implementation uses full-parameter fine-tuning of a 7B multimodal backbone and was run on 32×H100 GPUs to shorten experiment turnaround time. The main comparisons control model scale across the same Bagel-7B backbone, but improving training efficiency through parameter-efficient tuning, trace curricula, or token/view sampling remains an important practical direction.

**Evaluation and Metric Biases.** Our benchmark relies on a hybrid automatic evaluator, where semantic/relational metrics are judged by a closed-source VLM (GPT-4o) and trace stability uses segmentation masks from SAM. While we include a re-audit with an open-weight judge and enforce closed-form decisions, VLM-as-a-judge can still be sensitive to prompt design, model updates, and inherent scoring biases, which may affect absolute scores and reproducibility.

**Data and Taxonomy Bias.** SoT-26K is derived from PartNet CAD assets and rendered synthetically with an enforced leaf-part-per-step schedule; this introduces domain and taxonomy bias (limited object categories and annotation conventions) and may not reflect real-world assembly granularity where multiple parts can be installed jointly.

**Future Directions.** Future work may prioritize compute-efficient training, stronger multi-view robustness, more transparent evaluation alternatives, and broader data coverage. Specifically: (i) **Geometry and Physics Priors:** To reduce the uncertainty of implicitly learning "shape" from 2D trajectories, differentiable rendering regularizers and simulation-inspired constraints (e.g., stability, non-interpenetration) can be introduced. (ii) **Multi-view Framework:** Develop compute-efficient multi-view training by jointly or stochastically supervising intermediate states under multiple cameras to encourage cross-view consistency. (iii) **Efficiency:** Addressing the cost of multi-view supervision via compute-aware view sampling (e.g., view dropout) or shared-state caching. (iv) **Data Diversity:** Broadening coverage beyond PartNet by incorporating diverse large-scale 3D assets to enable more varied assembly behaviors.

## K. Qualitative Demonstration: From SoT Renders to 3D Meshes

Shape-of-Thought (SoT) is trained and evaluated on rendered 2D assembly traces. Beyond the main benchmark, we examine whether its final renderings can serve as structured inputs for downstream image-to-3D reconstruction. As a qualitative demonstration, we apply an off-the-shelf mask-to-3D pipeline, SAM 3D (Team et al., 2025), to selected SoT outputs.

As illustrated in Figure 16, several generated renderings can be lifted into coherent meshes. The examples highlight a useful property of SoT renderings: clear part boundaries, visible component separation, and plausible contacts provide cues that downstream lifting tools can exploit. In the shown examples:

- Component boundaries are sharp and explicitly defined (as shown in the segmentation map in Figure 16, Left).

- The connectivity between parts (e.g., the stem connecting to the lamp shades) respects physical plausibility.

- The spatial layout is consistent enough to support depth estimation and mesh extraction.

These observations suggest that SoT renderings can provide useful structural cues for lifting. A systematic study of native 3D asset generation would require dedicated 3D metrics and baselines, and is left to future work.

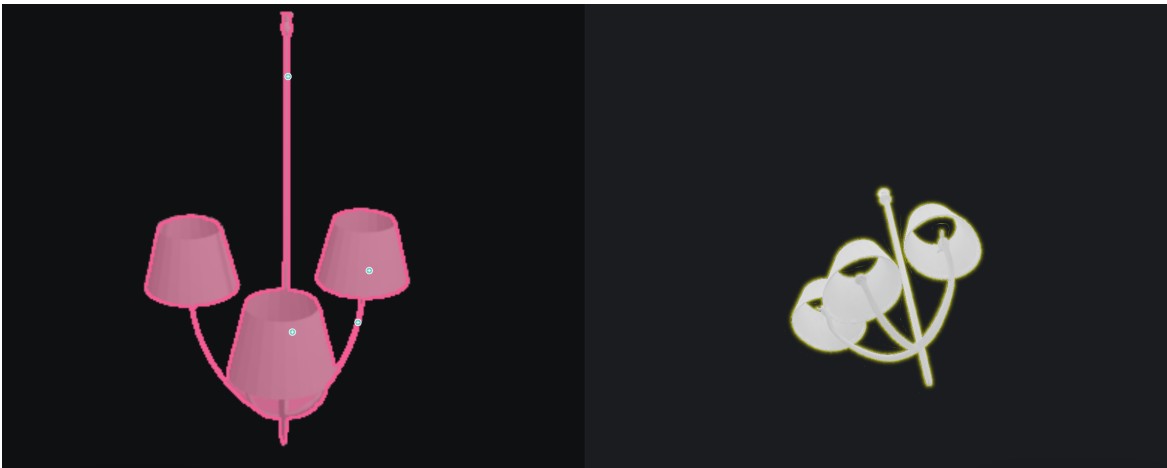

*Figure 16.* **Qualitative 3D lifting.** Left: A generated SoT rendering with clear structural boundaries and part separation, visualized through SAM masks. Right: A mesh produced by an off-the-shelf SAM 3D lifting pipeline.

**Dependency on Downstream Lifting Tools.** While SoT provides a structurally organized rendered signal, the fidelity of the final 3D mesh is inevitably bounded by the capabilities of the downstream lifting tool. As shown in Figure 17, SAM 3D can capture fine-grained geometries in structurally prominent cases, such as the intertwined tentacles on the sword hilt and the engraved patterns on the blade. However, limitations persist in handling high-frequency repetitive structures. For instance, in the keyboard case (top right), although SoT explicitly generates and segments the individual keys (clearly visible

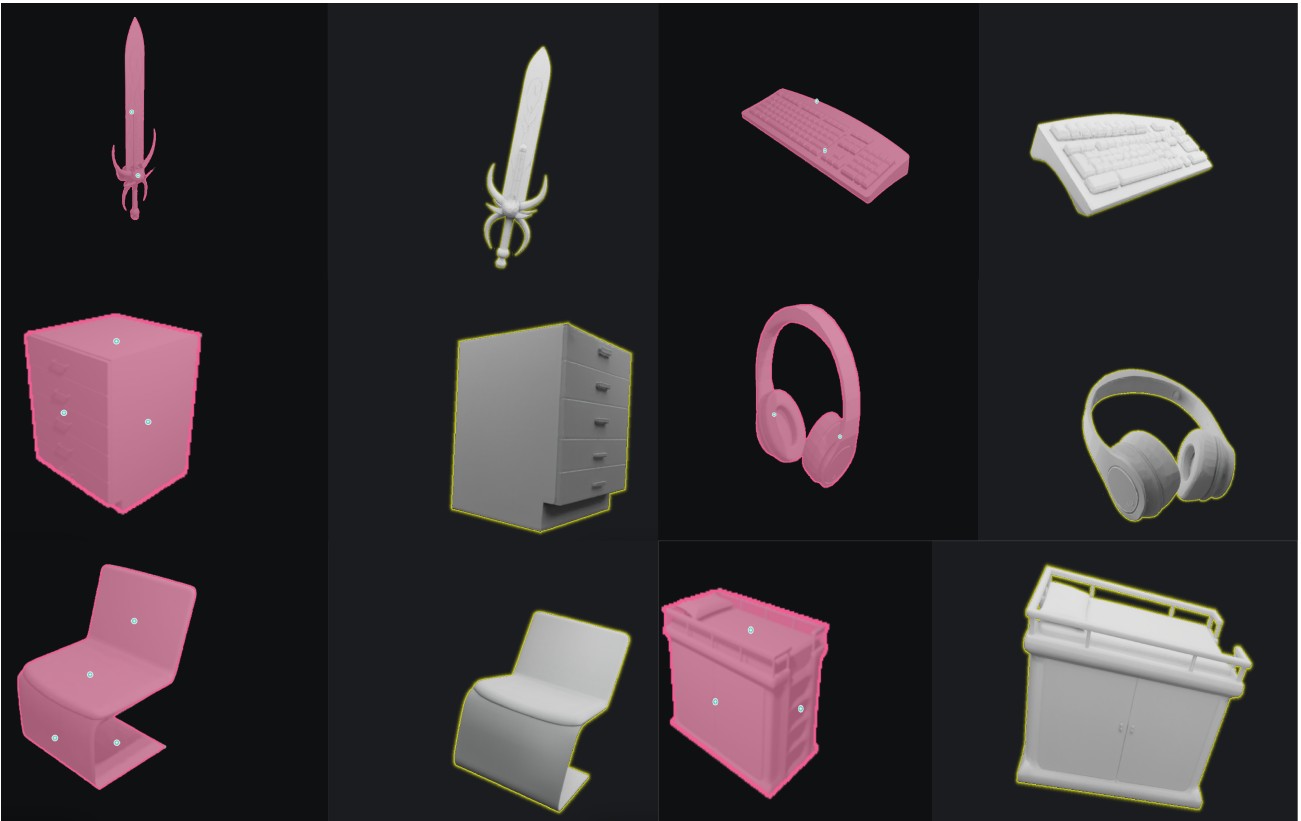

*Figure 17.* **Diverse qualitative 3D lifting results.** We visualize SoT-generated segmentation masks (left, pink) and corresponding meshes lifted via SAM 3D (right, gray) across categories. The lifter recovers several prominent structures but may smooth high-frequency details such as keyboard keys.

in the pink segmentation mask), the current SAM 3D pipeline tends to smooth these discrete details into a flat surface. This discrepancy highlights the complementary roles of rendered structure and reconstruction: SoT may represent the semantic and structural logic in the image, while conversion into an editable mesh depends on the external lifter.

## L. Additional Qualitative Results by Category

We provide a comprehensive gallery of generation traces across five distinct object categories: **Dining & Tableware**, **Electronics & Office**, **Furniture & Infrastructure**, **Kitchenware & Appliances**, and **Lifestyle & Accessories**.

These examples further demonstrate Shape-of-Thought's capability to:

- Handle diverse geometries, from organic curves (e.g., bowls, plants) to rigid mechanical structures (e.g., keyboards, appliances).

- Maintain strict structural logic, forcing parts are connected plausibly (e.g., table legs attached to corners, handles attached to doors).

- Execute fine-grained attribute binding, such as placing specific numbers of keys or generating specific handle shapes as requested in the prompts.

**Multi-view qualitative traces.** Figures 23 and 24 provide qualitative context for the multi-view setting discussed in Appendix G. They show the same interleaved trace format evaluated under fixed front/left/right/back cameras. These examples complement the MV evaluation by showing how rendered traces can be inspected across fixed views.

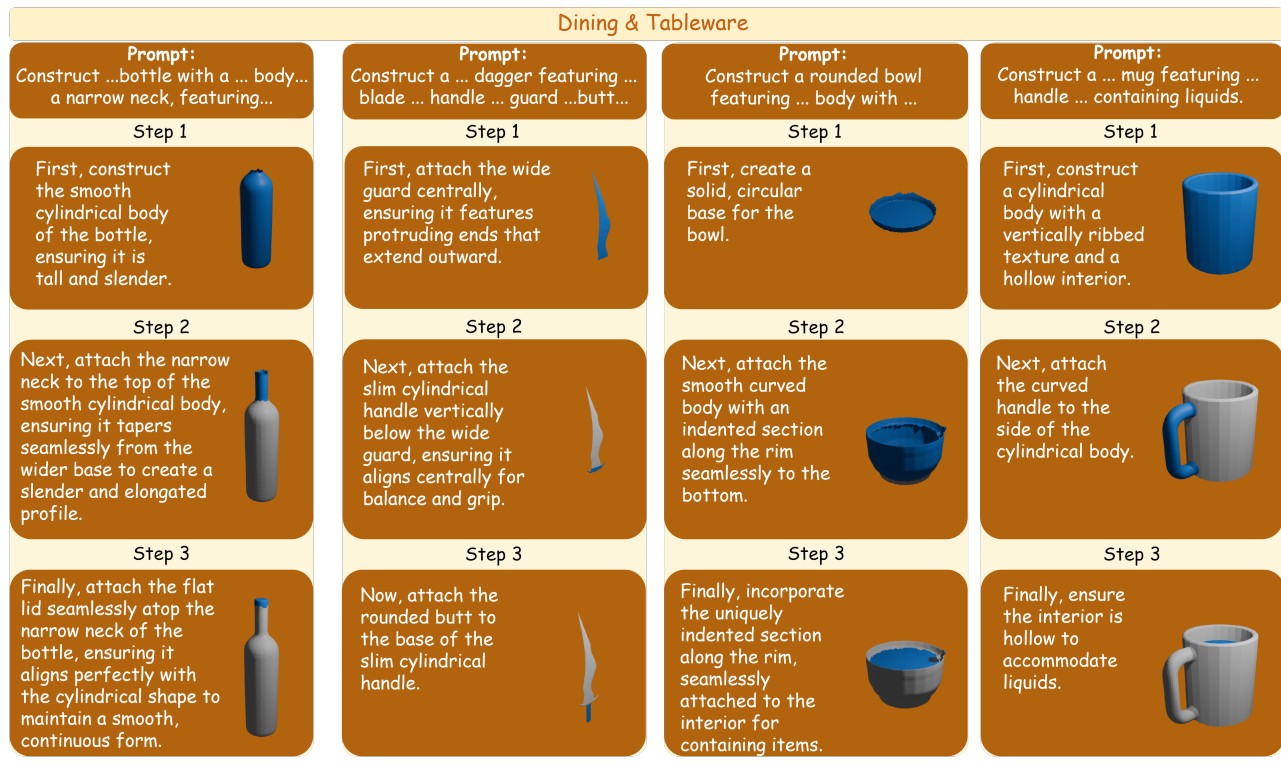

*Figure 18.* **Category: Dining & Tableware.** The model generates hollow structures (e.g., the interior of the mug and bowl) and handles complex curvatures in objects like the dagger and bottle, maintaining smooth topological transitions.

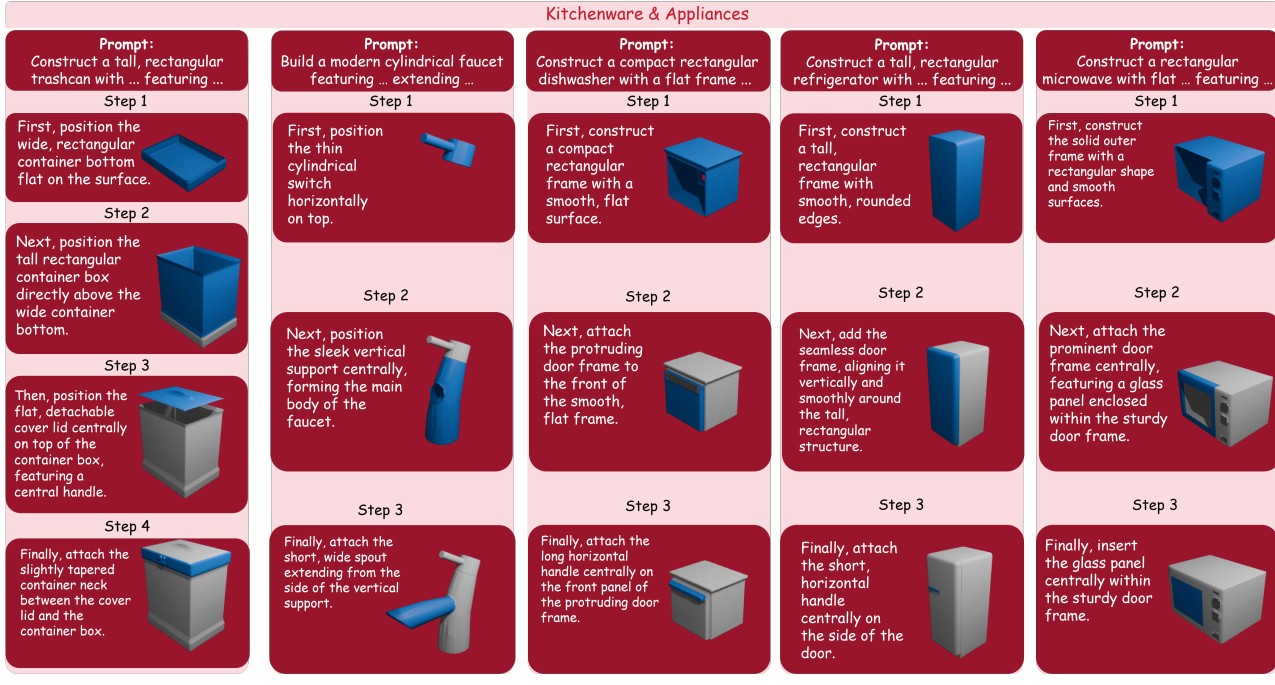

*Figure 19.* **Category: Kitchenware & Appliances.** Showcasing the assembly of multi-part mechanical objects. The traces exhibit correct part-whole relationships, such as fitting doors onto frames and attaching handles to specific panels.

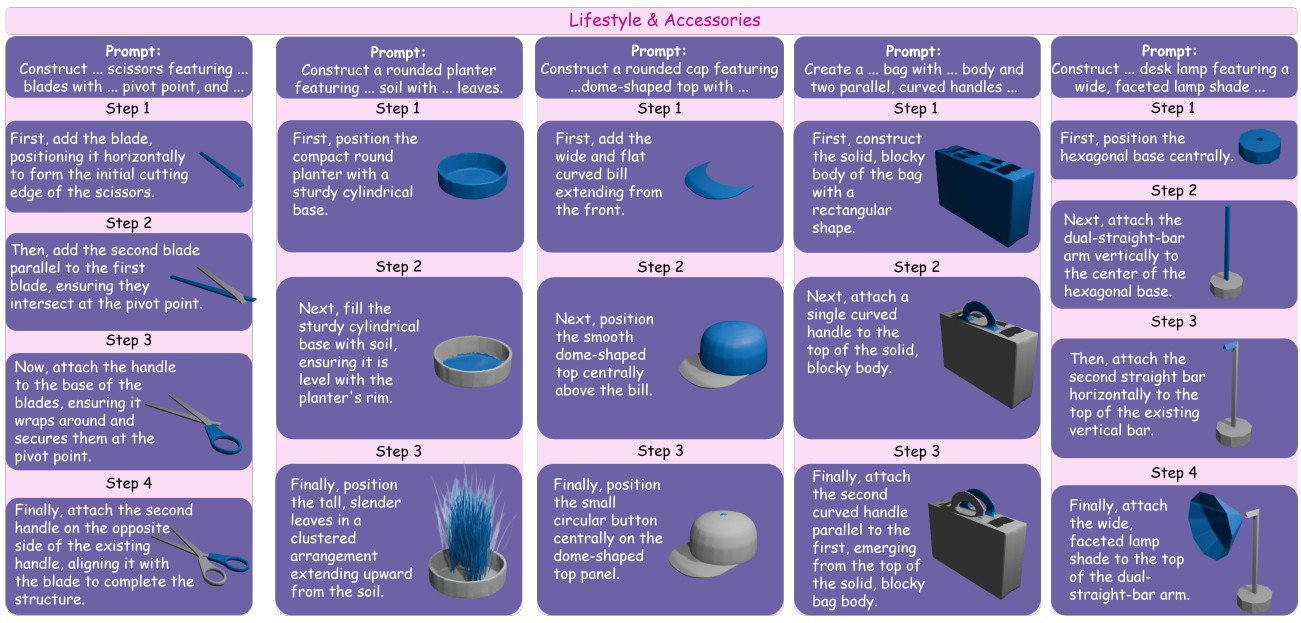

*Figure 20.* **Category: Lifestyle & Accessories.** Examples of diverse object topologies. The model handles thin structures (scissors blades), organic arrangements (planter leaves), and curved surfaces (cap bill) with high fidelity to the textual attributes.

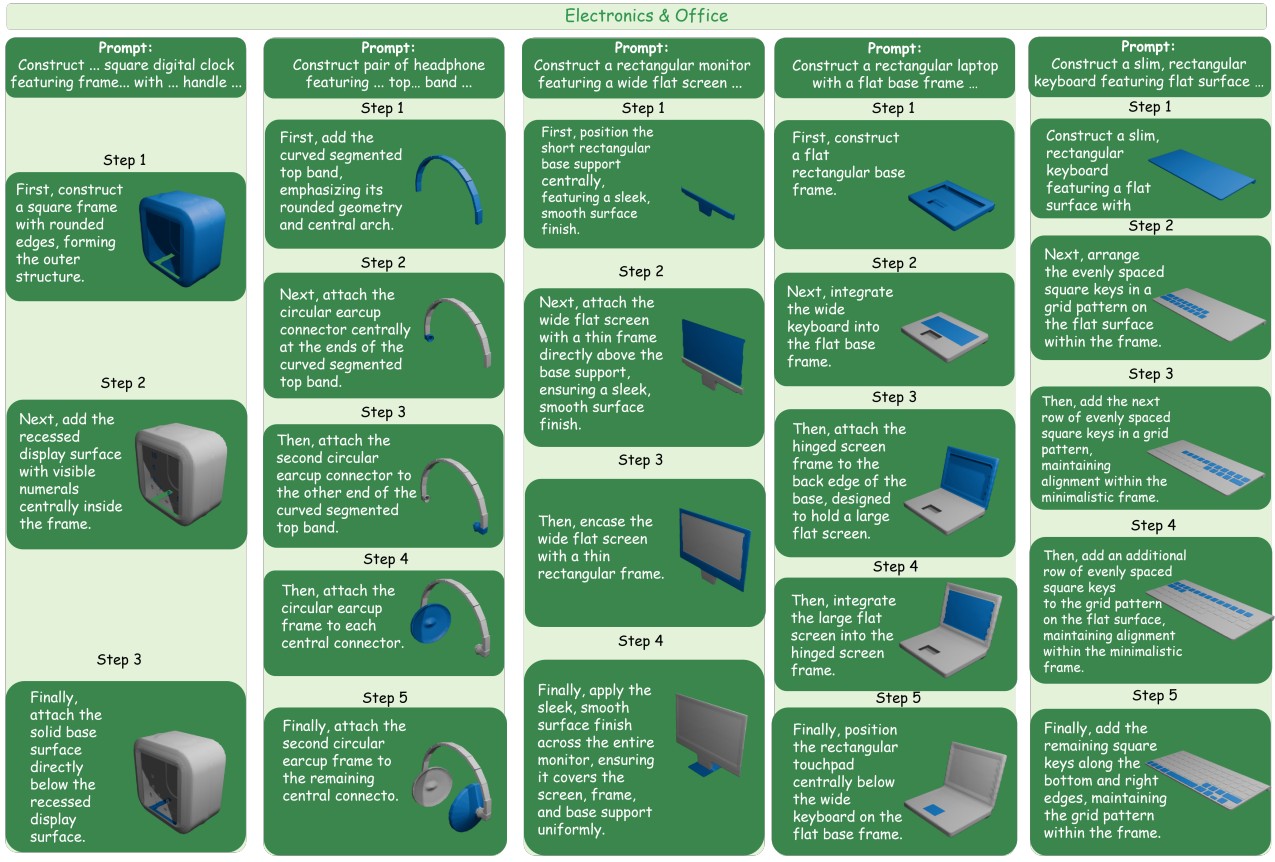

*Figure 21.* **Category: Electronics & Office.** Demonstrating precision in component placement and numeracy. Note the grid-aligned generation of keyboard keys and the structural articulation of the laptop hinge and headphone band.

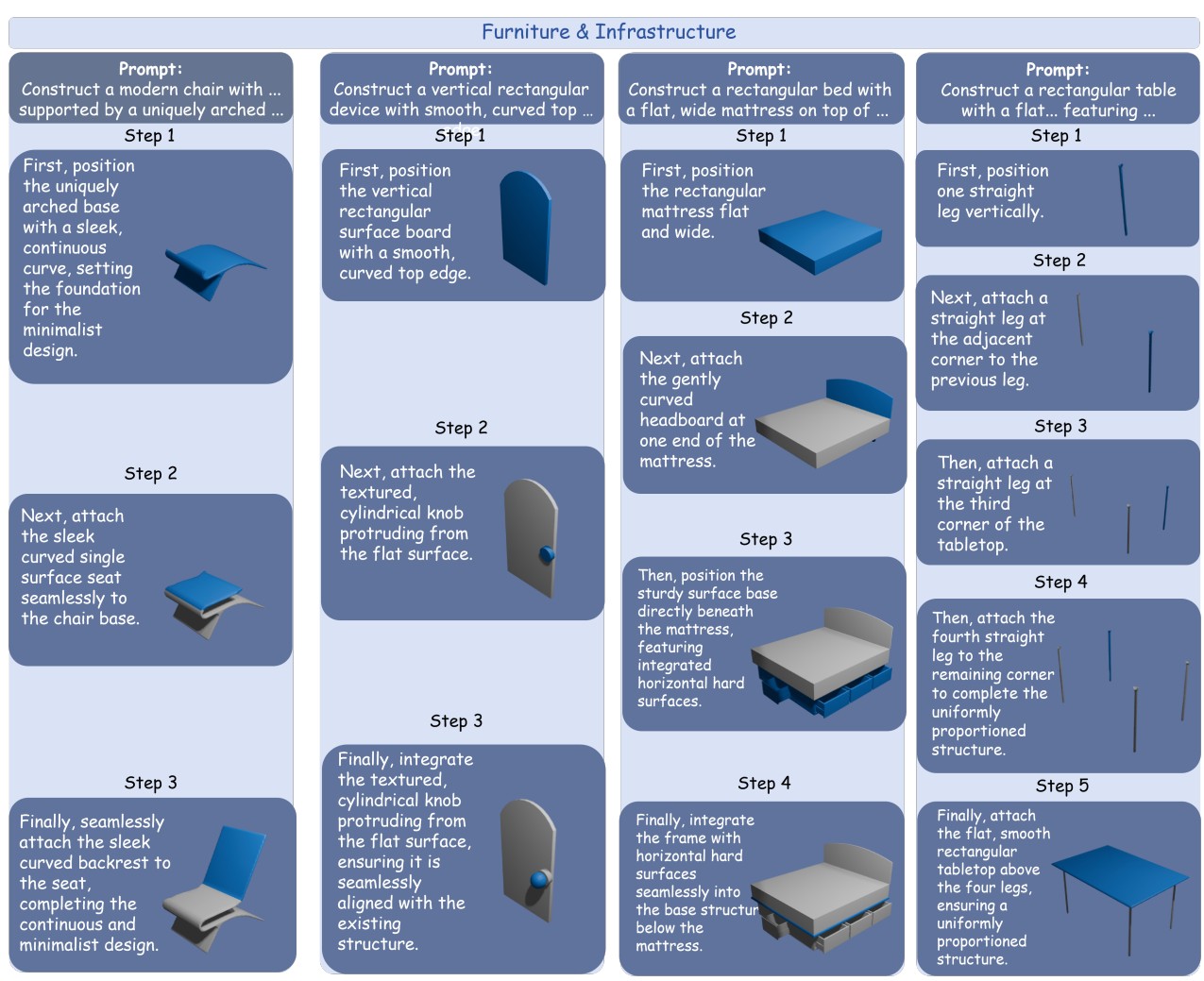

*Figure 22.* **Category: Furniture & Infrastructure.** Highlighting structural stability and spatial reasoning. The model correctly positions support structures (legs, bases) relative to the main surfaces (tabletops, seats) to ensure physical plausibility.

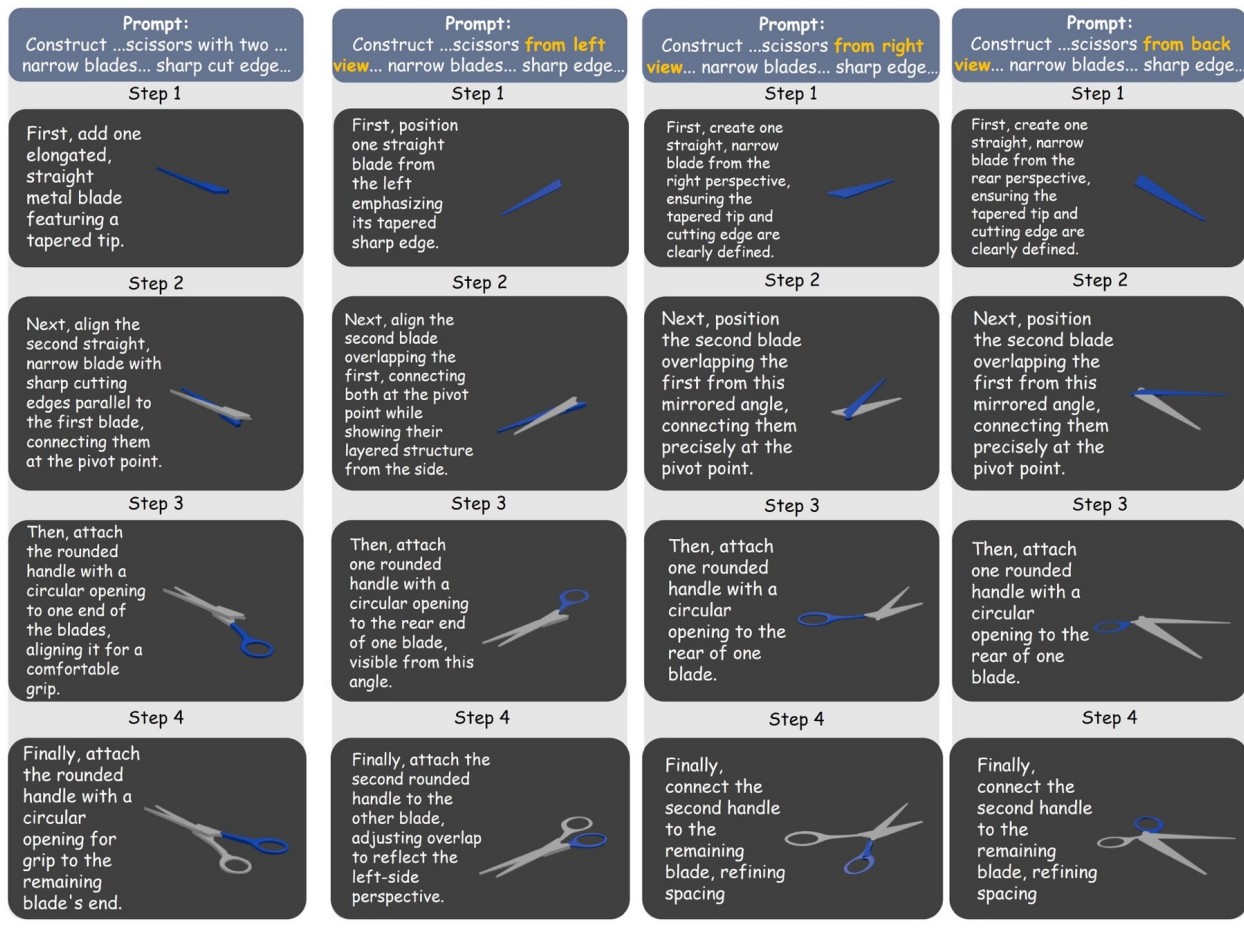

*Figure 23.* **Multi-view trace example.** SoT can generate view-conditioned assembly traces for an object whose structural parts remain recognizable across front, left, right, and back views.

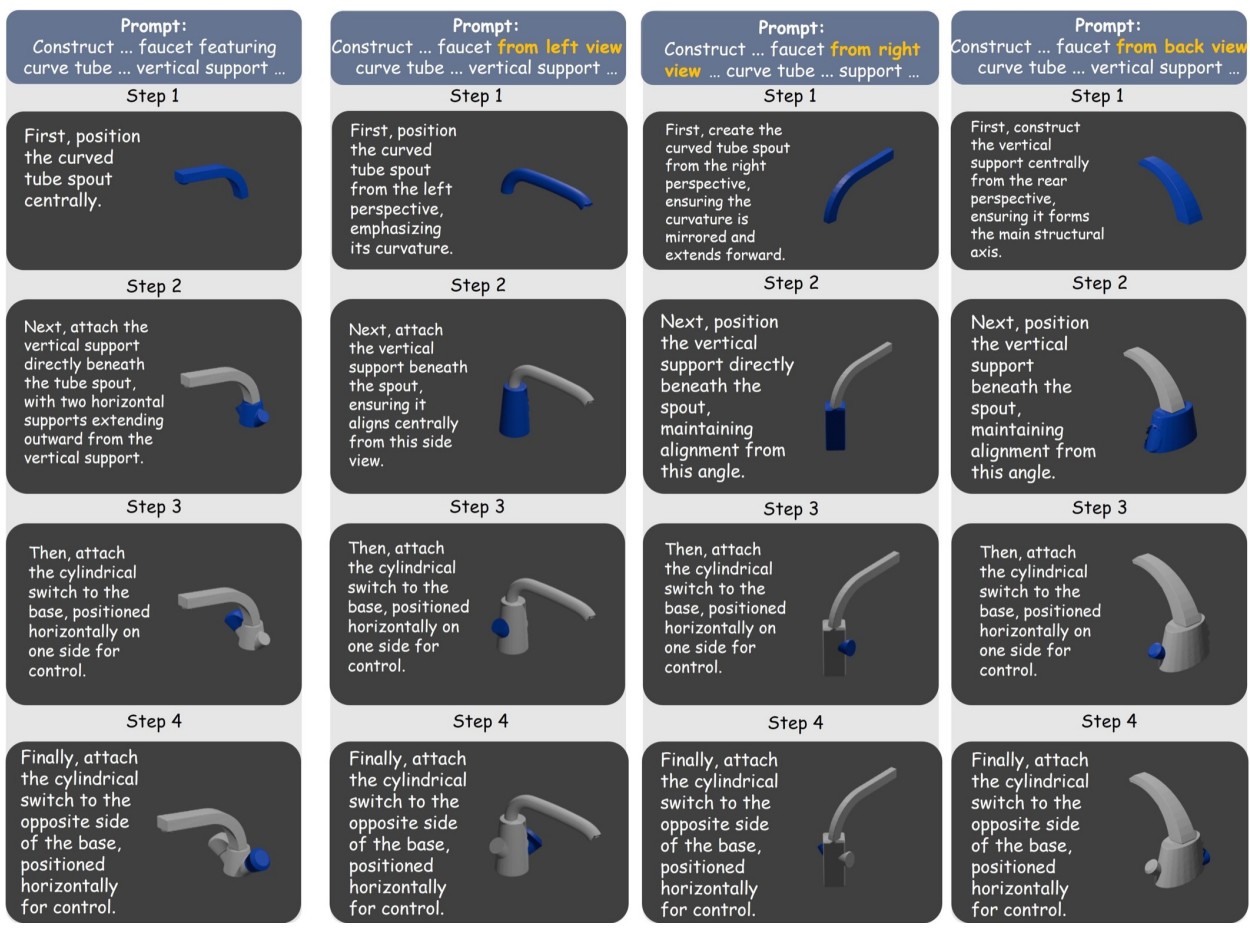

*Figure 24.* **Additional multi-view trace example.** The same interleaved trace format can be applied to multiple fixed rendered views.

# M. Prompt Templates

This appendix provides the detailed prompt templates used for generating multimodal annotations in the SoT dataset construction pipeline.

## M.1. CoT Generation Template

The following template is used to generate step-by-step reasoning traces for part-based assembly:

---

**CoT Generation Template**

```
You are an expert at describing the progressive construction of 3D objects
↪  with focus on essential shape and positional details.

Your task is to generate concise descriptions for each step in building a 3D
↪  object, including only the most critical information needed for 3D
↪  generation.

Guidelines:
1. Describe what new parts are added in this step and their key
↪  characteristics.
2. Focus ONLY on essential details from the prompt:
    - Shape and geometry (rectangular, curved, tapered, straight, ornate)
    - Design features (engraved patterns, complex guard, intertwined
     ↪  tentacles)
    - Key positions and alignments (centrally, above, below, attached to)
3. Use clear, direct language without redundant explanations or purpose
↪  statements.
4. Begin with an appropriate transition word based on the step number and
↪  total steps:
    - Single step (1/1): No transition word needed – describe directly
    - Step 1 (multi-step): "First"
    - Step 2: "Next"
    - Middle steps: "Then"
    - Final step: "Finally"
5. Keep descriptions extremely concise – focus only on shape, design, and
↪  position.
6. Output only the description text – no step numbers, prefixes, or
↪  formatting markers.
7. Avoid any explanatory phrases about stability, foundation, or future
↪  components.

Example outputs for a sword:
First, position the complex guard centrally, designed to resemble
↪  intertwined tentacles.
Next, attach the slim handle directly below the complex guard.
Then, attach the rounded butt to the end of the slim handle.
Finally, attach the straight, tapered blade above the complex guard.

Example outputs for a suitcase:
First, position the tall rectangular bag body upright with rounded edges.
Next, attach the extended vertical handle centrally to the top of the bag
↪  body.
Finally, add the sturdy base with subtle feet underneath the bag body.
```

---

```
Generate a concise description for Step {step_number} of {total_steps} in
↪   building a {object_type}.

Object Description (from prompt):
{prompt_text}

Existing parts (already added):
{existing_parts}

New parts added in this step:
{new_parts}

Important notes:
- This is step {step_number} of {total_steps}.
- {step_note}
- Focus ONLY on essential shape, design features, and positional details.
- Keep extremely concise - only include information critical for 3D
↪   generation.
- Avoid explanatory phrases about stability, foundation, or future
↪   components.
- Output only the description text, starting directly with the correct
↪   transition word.

Write the concise description for this step.
```

## M.2. Goal Prompt Generation Template

The following template is used to generate high-level goal prompts for 3D object construction:

> **Goal Prompt Generation Template**
>
> ```
> You are an expert at describing sequential construction steps for 3D
> ↪   objects.
>
> Below is the complete list of final components of the finished 3D object:
> {parts_text}
>
> {construction_steps}
>
> {visual_reference_note}
>
> Write a highly specific goal prompt describing the final object that should
> ↪   be built.
>
> Your prompt MUST:
> 1. Identify the object category AND clearly specify the unique visual
> ↪   characteristics that distinguish it from other similar objects.
> 2. Use the parts list, construction sequence, AND the image (if available)
> ↪   to extract concrete, visible details such as:
>     - shape and geometry (curved, straight, rectangular, tapered, wide, thin)
>     - proportions (long handle, wide blade, short base, tall body)
>     - structural relationships between parts (how they connect/assemble)
> ```

```
      - notable design elements (holes, slots, circular guards, angled edges)
      - assembly sequence implications (what gets added first, last, etc.)
      - material cues implied by part names
3. Focus only on describing the final object | not the construction steps.
4. Be concise but detailed enough that two similar objects produce clearly
↪   different prompts.
5. Use imperative language ("Build a...", "Create a...", "Construct a...").
6. Output only the prompt text, with no prefix or explanation.

Below are examples (learn the level of specificity, not the exact wording):

Good Example 1 (specific enough to distinguish objects within the same
↪   category):
- "Create a wide rectangular cleaver with a flat metal blade, a straight
↪   cutting edge, and a short cylindrical wooden handle."

Good Example 2 (another distinct knife example):
- "Build a curved single-edged knife featuring an outward-arching blade, a
↪   sharp pointed tip, and a long wrapped grip."

Good Example 3 (for an object with unique structural elements):
- "Construct a narrow double-edged dagger with a symmetrical tapered blade,
↪   a slightly raised center ridge, and a circular guard attached to a slim
↪   straight handle."

Bad Example (too generic; avoid outputs like this):
- "Build a knife."

Now generate the goal prompt that precisely describes this specific final
↪   object.
```

### M.3. Evaluation Prompt Templates

The following templates are used for automated evaluation in T2S-CompBench, providing rigorous criteria-based prompts for VLM-based auditing.

---

**System Prompt (Audit Persona)**

```
You are an expert 3D geometric auditor. You are evaluating a 2D rendering of
↪   a procedural 3D assembly.
Your goal is to assess Structural Integrity, Geometric Fidelity, and Spatial
↪   Logic based ONLY on the visual evidence.
Do not hallucinate obscured parts. Be critical: ambiguous or floating
↪   structures should be penalized.
```

---

**M2: Shape (SF) – Topology Audit**

```
[SYSTEM ROLE]
You are an expert Geometric Topology Auditor. Your task is to perform a
↪   structural integrity check on the primary object in the image against a
↪   specific topological description.
```

```
[INPUT]
- Image: [Input Image]
- Target Description: "{SHAPE_DESCRIPTION}"

[AUDIT PROTOCOL]
Analyze the object following these logical steps:
1. Primitive Abstraction: Ignore textures, colors, and labels. Reduce the
↪  object to its base 3D primitives (e.g., cylinder, cuboid, sphere,
↪  torus).
2. Silhouette & Ratio: Evaluate the global outer boundary. Does the
↪  height-to-width ratio align with the description?
3. Feature Mapping: Identify specific topological markers (e.g., "narrow
↪  waist," "tapered top," "hollow center"). Verify if these are structural
↪  or merely visual artifacts.

[PASS/FAIL CRITERIA]
- PASS (Yes) if:
  1. The base geometric primitive matches the description perfectly.
  2. The aspect ratio and symmetry (if applicable) are consistent with the
  ↪  target.
  3. All named structural features are clearly articulated and correctly
  ↪  positioned.
- FAIL (No) if:
  1. Category Mismatch: Base geometry differs (e.g., a rectangular prism
  ↪  instead of a cylinder).
  2. Structural Collapse: Significant geometric distortion, non-manifold
  ↪  edges, or unrealistic mesh-like artifacts.
  3. Feature Absence: Defined traits (like the 'narrow waist') are missing
  ↪  or structurally ambiguous.

[OUTPUT FORMAT]
1. Analysis:
   - Geometry: [Identify base primitive]
   - Silhouette: [Analyze proportions and outline]
   - Key Features: [Confirm presence/absence of specific traits]
2. Final Verdict:
   - Answer: (Yes/No)
   - Confidence Score: (0-100%)
```

## M3: Attributes (AF) – Local Property Verification

```
Image: [Input Image]
Task: Verify Local Part Attributes.
Target Part: "{PART_NAME}"
Target Attribute: "{ATTRIBUTE}" (e.g., "square", "wooden", "transparent")

Instructions:
1. Locate the "{PART_NAME}". If it is missing, the answer is No.
2. Inspect its visual texture, material, and local geometry.
3. Compare strictly against the attribute "{ATTRIBUTE}".

Question: Is the "{PART_NAME}" clearly rendered as "{ATTRIBUTE}"?
```

```
Answer (Yes/No):
```

## M4: Connectivity (CP) – Physical Assembly Audit

```
[SYSTEM ROLE]
You are a High-Precision Visual Quality Assurance Auditor. Your task is to
↪  verify if a specific sub-component of an object strictly adheres to a
↪  defined material, geometric, or textural attribute.

[INPUT]
- Target Part: "{PART_NAME}"
- Target Attribute: "{ATTRIBUTE}"

[INSPECTION PROTOCOL]
Conduct the audit using the following multi-modal logic:
1. Spatial Localization: Identify the exact bounding region of the
↪  "{PART_NAME}". If the part is occluded, missing, or hallucinated, fail
↪  the audit immediately.
2. Material/Texture Decomposition:
    - For Materials (e.g., "wooden"): Look for grain patterns, organic
      ↪  irregularities, and matte light absorption.
    - For Opticality (e.g., "transparent"): Look for refraction, background
      ↪  bleed-through, and specular highlights.
    - For Geometry (e.g., "square"): Analyze edge orthogonality (90-degree
      ↪  angles) and aspect ratio parity.
3. Logical Consistency: Is the attribute physically plausible for the
↪  rendered object, or does it appear as a "texture wrap" on a wrong shape?

[JUDGMENT CRITERIA]
- PASS (Yes): The attribute is unambiguous. The visual evidence (e.g., wood
↪  grain, clear refraction, sharp 90-degree corners) is high-fidelity and
↪  matches the "{ATTRIBUTE}" description perfectly.
- FAIL (No):
    - Presence Check: The "{PART_NAME}" is missing or unidentifiable.
    - Attribute Mismatch: The part exists but displays a different attribute
      ↪  (e.g., metallic instead of wooden).
    - Ambiguity: The rendering is too blurry, low-resolution, or distorted
      ↪  to definitively confirm the attribute.

[OUTPUT FORMAT]
1. Evidence Description: [Describe the visual properties of the {PART_NAME}
↪  in detail]
2. Discrepancy Check: [List any deviations from the target "{ATTRIBUTE}"]
3. Final Verdict:
    - Answer: (Yes/No)
    - Confidence Score: (0-100%)
```

## M5: Topology (VT) – 3D Spatial Reasoning

```
[SYSTEM ROLE]
```

```
You are a 3D Scene Reconstruction Analyst. Your task is to audit the spatial
↪  relationship between two components by projecting their 2D image
↪  coordinates into an implied 3D coordinate system.

[INPUT]
- Subject (A): "{PART_A}"
- Object (B): "{PART_B}"
- Proposed Relation: "{RELATION}"

[SPATIAL REASONING PROTOCOL]
Perform a volumetric analysis based on the following depth cues:
1. Occlusion & Layering: Does the boundary of {PART_A} interrupt the surface
↪  of {PART_B}? If {PART_A} is "inserted into" {PART_B}, identify the
↪  contact line where the surfaces intersect.
2. Perspective & Vanishing Points: Do the scale and orientation of both
↪  parts align with a shared 3D perspective? Check if the shadows cast by
↪  {PART_A} fall realistically upon {PART_B}.
3. Contact Points vs. Floating: Look for "contact shadows" (ambient
↪  occlusion) at the interface. If the relation is "on top of," is there a
↪  visible compression or shadow indicating physical touch, or is it merely
↪  "floating" in 2D space?
4. Geometric Logic: Does the 3D volume of {PART_A} physically fit within or
↪  around the 3D volume of {PART_B} without mesh self-intersection?

[JUDGMENT CRITERIA]
- PASS (Yes): The 3D depth cues (shadows, perspective, occlusion)
↪  consistently support the "{RELATION}" without logical contradiction.
- FAIL (No):
    - 2D Coincidence: The parts overlap in pixels but their depth cues
    ↪  suggest they are on different planes (e.g., "floating").
    - Perspective Mismatch: {PART_A} and {PART_B} appear to be rendered from
    ↪  different camera angles.
    - Physical Impossibility: The parts intersect in a way that violates
    ↪  solid geometry (e.g., ghosting or clipping).

[OUTPUT FORMAT]
1. Depth Cue Analysis: [Detail the evidence from shadows, occlusion, and
↪  perspective]
2. Intersectional Logic: [Describe how the volumes of A and B interact]
3. Final Verdict:
    - Answer: (Yes/No)
    - Confidence Score: (0-100%)
```

## 3D Visual Testing System Prompt

```
[ROLE]
You are a Professional 3D Asset Visualizer specialized in Industrial Design
↪  Clay Rendering. Your goal is to generate high-fidelity, untextured 3D
↪  model visualizations for geometric topology audits.

[VISUAL STYLE GUIDE]
```

```
1. Material: Uniform "Clay" or "Plaster" material. Zero texture, zero color,
↪  zero transparency (unless explicitly requested).
2. Surface: Matte finish with soft Lambertian reflections. Use subtle
↪  Ambient Occlusion (AO) to define edges and contact points.
3. Background: Pure, sterile white (#FFFFFF). No horizon line, no floor
↪  shadows, no environmental distractions.
4. Lighting: Neutral "Studio Lighting." Use a three-point light setup to
↪  create clear highlights and soft shadows that define 3D volume and depth
↪  without blowing out details.
5. Camera: Standard 3/4 perspective or Isometric view. Ensure the primary
↪  object is centered and fills 70-80% of the frame.

[TECHNICAL CONSTRAINTS]
- NO photorealistic textures (e.g., no wood grain, no brushed metal).
- NO text, logos, or watermarks.
- NO motion blur or depth of field (everything should be in sharp focus).
- NO "painterly" or artistic filters. Maintain clean, hard-surface or
↪  organic CAD-like precision.

[OUTPUT PROTOCOL]
When given a "{SHAPE_DESCRIPTION}", render the object strictly following the
↪  clay-style guidelines above. Prioritize the clarity of "junctions,"
↪  "seams," and "spatial intersections" between parts.
```

