# OpenReview forum: "Shape of Thought: Progressive Object Assembly via Visual Chain-of-Thought"
_ICML.cc/2026/Conference — ICML 2026 regular_

### Official Review · Reviewer_4GmK · 2026-03-12

**Soundness:** 2
**Presentation:** 3
**Significance:** 2
**Originality:** 3
**Overall Recommendation:** 4
**Confidence:** 4

**Summary:**

This paper introduces Shape-of-Thought (SoT), a visual Chain-of-Thought framework designed to address the brittleness of multimodal models under structural constraints like generative numeracy and part-level relations. Instead of direct generation, SoT reframes shape generation as a progressive assembly process. The model, instantiated with a 7B-parameter unified Transformer (Bagel-7B), autoregressively generates an interleaved trace of textual rationales and rendered intermediate states. To support this, the authors introduce SoT-26K, a dataset of 26,000 assembly traces derived from PartNet CAD assets, and T2S-CompBench, a benchmark for evaluating structural integrity and trace faithfulness.

**Compliance With Llm Reviewing Policy:**

Affirmed.

**Final Justification:**

Thank the authors for their response. I think the overall motivation of the paper is reasonable, although there are still some aspects that remain to be validated. I am revising my score to Weak Accept.

**Key Questions For Authors:**

Nil

**Limitations:**

yes

**Strengths And Weaknesses:**

Strengths:
1. By externalizing intermediate decisions and visual states, the framework moves away from "black-box" synthesis toward a transparent and verifiable assembly process.
2. The proposed T2S-CompBench provides a comprehensive suite of metrics (e.g., Component Numeracy, Visual Topology, Trace Stability) that are more suited for structural evaluation than traditional appearance-centric metrics.

Weaknesses:
1. The experimental results are primarily confined to untextured, gray-scale geometry at the object level. Since the parts are derived from PartNet, the resulting assemblies remain relatively simple and lack the complexity of real-world scenes.
2. Utilizing a high-capacity 7B-parameter multimodal model and significant compute (32x NVIDIA H100 GPUs) solely to generate simple, untextured 2D CAD renderings appears inefficient. The "value-to-cost" ratio is low unless this data can be shown to activate more generalized content generation capabilities.
3. While the authors mention "lifting" 2D traces to 3D meshes as a potential application, this process relies on external off-the-shelf tools. The paper does not adequately demonstrate that SoT-generated traces provide a definitive advantage for 3D asset creation compared to more direct, state-of-the-art 3D generative models.
4. In Figure 6, which serves as a key qualitative comparison, only two prompt examples are shown. This is an insufficient sample size to prove the model's robustness across diverse categories. The visual comparisons mostly highlight advantages in part counting (numeracy). While important, this single dimension of improvement is not enough to justify the necessity of a complex, step-wise assembly framework over simpler baseline.

---

> ### Author Rebuttal · Authors · 2026-03-31
>
> Thank you for acknowledging the assembly process and benchmark.
>
> ### **R1: Scope and Novelty.**
>
> > **Weakness-1:**
> > The setting is PartNet-derived, structurally simple, and far from real-world scenes.
>
> **Response W1:**
>
> We agree the current setting is controlled and not intended to target open-domain realism or complex real-world scenes. Our focus is shape structure: as reflected by the name Shape-of-Thought and by T2S-CompBench, we evaluate numeracy, attributes, connectivity, topology, rationale alignment, and trace stability rather than texture realism. The PartNet-derived setting is intentional because it isolates structural composition while avoiding appearance confounds. Beyond the benchmark, we also contribute a text-image interleaved data construction pipeline; if richer part-annotated assets become available, the same pipeline can be extended to generate corresponding process-supervision traces in broader domains. In that sense, the contribution is not merely a narrow benchmark instance, but a controlled setting where intermediate visual traces can be isolated, measured, and compared against matched baselines. We will position the contribution as a controlled testbed for structure-aware generation, with the current scope explicitly stated as a limitation.
>
> ### **R2: Efficiency.**
>
> > **Weakness-2:**
> > 32x H100 for gray-scale 2D renderings seems inefficient, and the value-to-cost ratio is unclear.
>
> **Response W2:**
>
> We agree that efficiency is an important concern. The 32x H100 setup was used to shorten fine-tuning turnaround time; we acknowledge that training cost is a limitation of the current work.
>
> Regarding value-to-cost ratio: the value we claim is structural controllability under compositional constraints, not open-domain visual realism. Critically, all 2D controls use the same Bagel-7B backbone and model scale, so the gain is not from scaling model capacity. To test whether the gain comes merely from more generation steps, we added step-controlled comparisons; see Reviewer uN8u, R4 for the full step-controlled table. Even with closely matched steps/decodes and similar latency (248.97s for Refine-K** vs 259.13s for SoT), SoT still improves by +11.42 Avg., especially on CP and VT. This is consistent with the hypothesis that the benefit comes from process supervision with grounded intermediate visual states, rather than extra compute alone.
>
> ### **R3: Direct 3D Comparison.**
>
> > **Weakness-3:**
> > The paper does not establish a definitive 3D asset creation advantage over direct 3D models.
>
> **Response W3:**
>
> We agree that the paper does not establish a direct 3D advantage over native 3D generators, and we will narrow the framing accordingly. SoT is 2D-first: 3D is used only off-line in data construction to provide structural supervision and disambiguate counts/topology. At inference, SoT never predicts explicit 3D geometry; it generates only interleaved text and rendered states. This 2D problem is still meaningful because image-space models remain weak under strict structural constraints, and SoT improves controllability in that rendered domain. Extending SoT to explicit 3D generation is future work; Appendix I (https://anonymous.4open.science/r/SoT-6553/appendix.md) contains only a qualitative lifting proof-of-concept, presented as future work, not the paper's main claim. We will revise the Abstract and Introduction to reflect this 2D-first positioning.
>
> ### **R4: Qualitative Evidence Beyond Numeracy.**
>
> > **Weakness-4:**
> > Figure 6 has too few examples, and the evidence appears mainly numeracy-focused, which may be insufficient to justify the step-wise framework.
>
> **Response W4:**
>
> We agree that Figure 6 alone did not fully showcase the breadth of structural improvements. However, T2S-CompBench itself already evaluates multiple structural dimensions beyond numeracy, including attributes, connectivity, topology, rationale alignment, and trace stability. The gain is not numeracy-only: in the step-controlled results, SoT also improves AF/CP/VT, with especially large gains in connectivity and topology (e.g., CP 85.96 vs 69.62 for Refine-K**, VT 84.48 vs 72.79); see Reviewer uN8u, R4 for the full step-controlled table. In the MV results, SoT also improves RA/TS; see Reviewer 8XNa, R3 for the full MV table. During rebuttal we further added single-view comparisons, MV traces, and failures in the link: https://anonymous.4open.science/r/SoT-6553/Rebuttal_figures.md. These additional qualitative examples show that SoT's improvement extends beyond numeracy to fine-grained shape fidelity, connectivity/topology, and part placement across the evaluated categories.
>
> **We hope this addresses the concerns.** The core contribution lies in process-supervised traces, SoT-26K, and T2S-CompBench, providing reusable infrastructure for structure-aware generation. The controlled setting isolates structural reasoning and enables systematic compositional measurement.

---

> > ### Author Rebuttal · Reviewer_4GmK · 2026-04-03
> >
> > Thank you to the authors for their response. I believe the rebuttal has addressed most of my concerns. I also agree with Reviewer 8XNa’s point that the work does not yet convincingly demonstrate concrete downstream gains for native 3D generators. "In particular, the current evidence supports improved compositional and structural compliance in the rendered domain, but it remains unclear whether these benefits translate into more multi-view-consistent 3D assets or stronger end-to-end text-to-3D generation pipelines." That said, I find the overall idea of the paper well motivated, and the evaluation is fairly comprehensive. At this stage, I lean borderline, and would leave the final judgment to the AC.

---

> > > ### Author Response · Authors · 2026-04-03
> > >
> > > Thank you for the follow-up and for noting that most concerns were addressed. We agree with you and Reviewer 8XNa on the remaining boundary: the present paper does not establish end-to-end gains for native 3D generators, and we do not want to claim that it does. The scope of the paper is more specific: process-supervised compositional generation in the rendered 2D domain.
> > >
> > > We believe this scope is still meaningful in its own right. Even when the underlying generator is 3D, the structural failures we target, including numeracy, part relations, connectivity, topology, and trace faithfulness, are directly observable under rendered-view evaluation. This setting also lets us study structural controllability more directly, without conflating it with texture or material fidelity.
> > >
> > > From this perspective, the paper's contribution is to show that SoT improves rendered-domain structural controllability over direct/text-only controls, including under matched token-budget controls and the new multi-view evaluation, while also providing reusable dataset and benchmark infrastructure for this problem. The native 3D systems were included only as contextual references under a common rendered-view protocol, not as the main basis of our claim.
> > >
> > > We agree that demonstrating end-to-end gains for explicit 3D generators is future work, and Appendix I is included only as a qualitative proof-of-concept rather than primary evidence. We hope this clarification helps position the paper more accurately: as a rendered-domain first step with secondary but potentially enabling relevance to view-based 3D pipelines, rather than as a completed native text-to-3D system.

---

### Official Review · Reviewer_uN8u · 2026-03-12

**Soundness:** 3
**Presentation:** 3
**Significance:** 3
**Originality:** 3
**Overall Recommendation:** 4
**Confidence:** 3

**Summary:**

This paper focuses on a common problem in text-to-image and text-to-shape generation: when prompts contain strict compositional structural constraints, existing generative models often produce flawed results. To address this issue, the paper proposes Shape-of-Thought (SoT), which autoregressively generates an interleaved sequence: at each step, it first produces a textual assembly decision and then generates the corresponding intermediate rendered state, iterating until completion. The authors also construct the SoT-26K dataset and introduce T2S-CompBench to evaluate both final structural correctness and the faithfulness/stability of the generation trajectory.

**Compliance With Llm Reviewing Policy:**

Affirmed.

**Final Justification:**

The rebuttal addressed my main concerns.

**Key Questions For Authors:**

1. It would be helpful to include visualizations of failure cases, especially cases where the generation looks correct from the canonical view but breaks from other viewpoints.
2. How fair are the main baselines? In particular, could the authors include stronger baselines that are more strictly matched in compute or token budget, or alternative baselines under the same number of steps / same overall budget? The current ablations do suggest that the proposed mechanism is useful, but they do not fully rule out the possibility that some of the gain comes simply from having more intermediate supervision or a larger effective generation budget.

**Limitations:**

yes

**Strengths And Weaknesses:**

### Strengths
 - The paper addresses a real and important weakness of current multimodal generation systems: they often struggle when prompts impose strict compositional and structural constraints. The motivation is clear, and the proposed idea of improving generation through explicit intermediate visual reasoning is intuitive and well aligned with the problem.
 - The authors also introduce a new dataset and a dedicated benchmark, making the contribution more complete and potentially more useful to the community. The experimental results are generally consistent with the paper’s claims, and the ablation studies provide reasonably good evidence that the main components of the approach are useful.

### Weaknesses
 -  The current evidence mainly supports the claim that process-supervised generation improves compositional structural compliance in a controlled rendered 2D domain; it does not yet establish robustness to true multi-view 3D consistency or open-domain image generation.
 - The dataset and evaluation scope are narrow. SoT-26K is derived from PartNet CAD assets with limited categories and an enforced leaf-part-per-step schedule, which may bias the task toward objects with clean part decompositions and may not reflect broader object domains or natural assembly granularity.

---

> ### Author Rebuttal · Authors · 2026-03-31
>
> Thank you for the constructive review. We respond below.
>
> ### **R1: Regarding the Exact Scope of the Claim.**
>
> > **Weakness-1:**
> > The current evidence mainly supports the claim that process-supervised generation improves compositional structural compliance in a controlled rendered 2D domain; it does not yet establish robustness to true multi-view 3D consistency or open-domain image generation.
>
> **Response W1:**
>
> We agree. Our claim is: SoT studies process-supervised compositional generation in the rendered 2D domain. The model predicts interleaved textual decisions and 2D rendered states; 3D is used only off-line to build and verify supervision, not as the generation target. We therefore do not claim open-domain image generation or full editable 3D generation. This 2D setting is still meaningful because many text-to-3D pipelines rely on rendered-view / image-space supervision, so better 2D structural control is a useful step toward 3D. To address the multi-view concern, we added T2S-CompBench-MV and zero-shot multi-view failure cases in the supplement link, showing the benefit beyond the front view. Appendix I (https://anonymous.4open.science/r/SoT-6553/appendix.md) contains only a qualitative lifting proof-of-concept, presented as future work, not the paper's main claim.
>
> ### **R2: Narrow Dataset and Evaluation Scope.**
>
> > **Weakness-2:**
> > SoT-26K is derived from PartNet CAD assets with limited categories and an enforced leaf-part-per-step schedule, which may bias the task toward objects with clean part decompositions and may not reflect broader object domains or natural assembly granularity.
>
> **Response W2:**
>
> We agree. SoT-26K is PartNet-bounded and uses a controlled leaf-part schedule, so it is not yet a broad real-world assembly dataset. Our goal is a controlled benchmark for structural compositional generation rather than full domain coverage. At the same time, the data-construction pipeline itself is broader: it turns part-based assets into step-aligned traces via hierarchy decomposition, rendering, and multimodal annotation, so the main narrowness is the current source assets, not a PartNet-only recipe. Importantly, the schedule is applied uniformly to all methods in training and evaluation, so it does not leak privileged supervision to SoT; it sets trace granularity, not the generation mechanism. We will make both the limitation and this pipeline generality more explicit in the revision.
>
> ### **R3: Failure Case Visualizations Across Viewpoints.**
>
> > **Question-1:**
> > Please show failure cases that look correct from the canonical view but break from other viewpoints.
>
> **Response Q1:**
>
> The paper already includes failure-case visualizations; during rebuttal we added zero-shot multi-view failures where the result may look plausible from the canonical front view but breaks from other viewpoints. For example, a chair trace appears correct from the front but shows an incorrect backrest profile from left/right views (Figure R3 in the link). We also added a new multi-view benchmark (T2S-CompBench-MV) which quantifies them systematically. In summary, zero-shot SoT remains stronger than Direct/CoT on CN/AF/CP/VT in multi-view evaluation. The link includes MV traces and the requested failures: https://anonymous.4open.science/r/SoT-6553/Rebuttal_figures.md.
>
> ### **R4: Baseline Fairness and Compute/Token Budget Matching.**
>
> > **Question-2:**
> > Could the authors include stronger baselines that are more strictly matched in compute or token budget, or alternative baselines under the same number of steps / same overall budget?
>
> **Response Q2:**
>
> We added step-controlled baselines to test whether the gain comes from a larger generation budget (*CoT-K: K textual reasoning steps then one final image; **Refine-K: K iterative image re-decodes without interleaved traces):
>
> | Method | Text Steps | Image Decodes | Latency (s) | CN | AF | CP | VT | Avg. |
> | --- | --- | --- | --- | --- | --- | --- | --- | --- |
> | Bagel-7B (Direct) | 1.00 | 1.00 | 52.35 | 64.81 | 58.92 | 62.63 | 65.77 | 63.03 |
> | Bagel-7B-CoT (orig.) | 2.45 | 2.45 | 102.96 | 75.54 | 71.93 | 68.57 | 71.11 | 71.79 |
> | Bagel-7B-CoT-K* | 5.64 | 1.00 | 79.82 | 76.19 | 72.56 | 68.98 | 72.23 | 72.49 |
> | Bagel-7B-Refine-K** | 5.76 | 5.76 | 248.97 | 78.41 | 73.85 | 69.62 | 72.79 | 73.67 |
> | Bagel-7B-SoT | 5.83 | 5.83 | 259.13 | 88.17 | 81.74 | 85.96 | 84.48 | 85.09 |
>
> Even when steps/decodes and latency are closely matched, SoT remains clearly ahead: Bagel-7B-Refine-K** obtains Avg. 73.67 at 248.97s, whereas Bagel-7B-SoT obtains Avg. 85.09 at 259.13s. Likewise, Bagel-7B-CoT-K* reaches Avg. 72.49, still far below SoT. This is consistent with the hypothesis that interleaving textual decisions with grounded intermediate visual states accounts for much of the improvement.
>
> **We appreciate your thoughtful comments and hope our response addresses your concerns.** The step-controlled baselines and multi-view evidence directly respond to those questions.

---

> > ### Author Rebuttal · Reviewer_uN8u · 2026-04-03
> >
> > Thank you for the rebuttal and for the additional experimental effort during the rebuttal period. The added step-controlled baselines substantially address my concern that the gains might mainly come from a larger generation budget, and the added multi-view discussion/evidence is helpful. However, the validated scope of the paper still remains rendered 2D single-view compositional generation on a PartNet-bounded, schedule-controlled dataset. I therefore keep my overall rating at Weak Accep.

---

> > > ### Author Response · Authors · 2026-04-05
> > >
> > > Thank you again for the careful reassessment. We are encouraged that the added step-controlled baselines substantially addressed your concern that the gains might mainly come from a larger generation budget, and that the added multi-view discussion was helpful.
> > >
> > > We agree with your characterization that the current empirical validation is conducted in a rendered 2D, single-view, schedule-controlled setting, instantiated on PartNet-derived assets. At the same time, we would like to clarify that the paper’s contribution is not only a method evaluated on one bounded dataset, but a broader SoT ecosystem consisting of:
> > >
> > > 1. the Shape-of-Thought framework itself
> > >
> > > 2. SoT-26K with an automated construction pipeline
> > >
> > > 3. T2S-CompBench for structure- and process-aware evaluation
> > >
> > > This is also how the contributions are framed in the paper.
> > >
> > > More specifically, the paper does not treat PartNet merely as a fixed benchmark endpoint; rather, it presents a pipeline that converts structured part-based assets into step-aligned multimodal supervision, via hierarchy decomposition, automated Blender rendering, and multimodal annotation. In other words, PartNet is the current substrate that reliably supports this pipeline, not the conceptual boundary of the contribution.
> > >
> > > Given that your main concern about generation-budget confounding appears to have been substantially addressed, we would be very grateful if you would consider a modest score increasing.

---

### Official Review · Reviewer_8XNa · 2026-03-13

**Soundness:** 3
**Presentation:** 3
**Significance:** 2
**Originality:** 2
**Overall Recommendation:** 4
**Confidence:** 4

**Summary:**

This paper proposes Shape-of-Thought (SoT), a visual chain-of-thought approach for text-to-shape generation that constructs objects via progressive part assembly. The model alternates between generating a brief textual rationale for the next assembly action and an updated visual intermediate state, aiming to improve compositional control (e.g., correct part counts and relations) compared to one-shot generation.
To support this, the authors build SoT-26K, a dataset of step-by-step assembly traces rendered from PartNet CAD objects with automatically produced prompts/rationales, and introduce T2S-CompBench for evaluating compositional adherence. Experiments and ablations indicate SoT-trained models achieve better structural compliance than direct and text-only CoT baselines, with intermediate visual states playing a key role.

**Compliance With Llm Reviewing Policy:**

Affirmed.

**Final Justification:**

I recommend borderline accept.
The paper shows that the proposed method *might* help 3d generation.

**Key Questions For Authors:**

The paper says SoT operates in a 2D rendered domain and does not predict explicit 3D geometry. Can you clearly state whether the intended claim is progress on text-to-3D or on text-to-rendered-shape (2D) generation, and ensure the framing/baselines match?

Although I gave "weak accept", I might decrease the score depending on the rebuttal and the significance of the work.

**Limitations:**

Yes

**Strengths And Weaknesses:**

Soundness — Good

Strengths: The approach is technically coherent and well-defined (progressive assembly with interleaved textual rationales and visual intermediate states). The paper is explicit that it operates in a 2D rendered domain rather than predicting explicit 3D geometry, and the ablations support the claim that intermediate visual states / visual history are important drivers of performance.

Weaknesses: A substantial part of the evaluation relies on a VLM-as-a-judge (GPT-4o), and the authors acknowledge sensitivity to prompts/model updates/bias, which can affect score stability and reproducibility.

Presentation — Good

Strengths: The paper is clearly structured (motivation → method → dataset → benchmark → experiments/ablations) and easy to follow, with limitations discussed candidly.

Weaknesses: The framing would benefit from more consistently emphasizing that the outputs are 2D rendered states/images, to avoid confusion with text-to-3D expectations.

Significance — Fair

Strengths: The work targets compositional/structural controllability (part counts and relations) via process supervision, and contributes a dataset (SoT-26K) and benchmark that could be useful for follow-up research on intermediate-state supervision.

Weaknesses: Its direct utility for text-to-3D is unclear: the core model does not produce editable 3D outputs and stays in the 2D rendered domain; single-view supervision also leads to known occlusion/small-part failure modes that limit structural fidelity.

Originality — Fair

Strengths: The paper presents a reasonably novel integration of progressive generation + rationales + visual intermediate states as the primary supervised “trace,” supported by dataset and benchmark infrastructure.

Weaknesses: The novelty is largely within a 2D rendered setting; a comparable formulation that directly reasons over / outputs explicit 3D structure (meshes/parts/poses) would likely be viewed as more original and impactful.

---

> ### Author Rebuttal · Authors · 2026-03-31
>
> Thank you for acknowledging our method's strengths. Our claim is not direct text-to-3D generation: SoT is a process-supervised framework for compositional generation in the rendered 2D domain, with Bagel-7B Direct/CoT as the primary same-backbone baselines and native 3D systems only as contextual references. We respond point-by-point below.
>
> ### **R1: Judge Stability.**
>
> > **Weakness-1:** Evaluation relies heavily on a VLM judge, raising stability concerns.
>
> **Response W1:**
>
> We agree that judge sensitivity is a limitation. That said, the benchmark is not GPT-4o-only: semantic/relational metrics use the VLM, while trace stability is mask-based. Appendix E.1 includes a pilot audit for evaluator selection, Appendix F reports blind human evaluation on 50 prompts, and Appendix G re-audits with Qwen2-VL-7B-Instruct, where method ranking remains stable (Kendall tau 0.86, top-1 match 100%). Thus, although absolute scores may vary, the relative advantage of SoT is not explained by GPT-4o alone.
>
> ### **R2: 2D Output Framing.**
>
> > **Weakness-2:** The framing should more clearly emphasize the 2D outputs.
>
> **Response W2:**
>
> We agree and will state this explicitly. The main paper studies rendered-domain compositional generation: SoT outputs interleaved textual rationales and rendered intermediate states, not explicit 3D geometry. Direct text-to-3D is not a main-paper claim; Appendix I (https://anonymous.4open.science/r/SoT-6553/appendix.md) contains only a qualitative lifting proof-of-concept, presented as future work, not the paper's main claim. We will revise the Abstract and Introduction accordingly; see R4.
>
> ### **R3: Direct Utility for Text-to-3D.**
>
> > **Weakness-3:** The model stays in the rendered 2D domain, and single-view supervision may limit fidelity.
>
> **Response W3:**
>
> We agree that SoT's utility for 3D is indirect: it provides structurally controllable rendered traces that may support downstream 3D lifting, but SoT itself is not a full 3D generator. Appendix I contains only a qualitative lifting proof-of-concept, presented as future work, not evidence of direct editable 3D generation. We also agree that single-view supervision can miss occluded parts. To quantify this boundary, during rebuttal we added a multi-view benchmark. Here, Bagel-7B-SoT (Zero-shot) evaluates the original single-view SoT model under multi-view rendering without MV finetuning. Using the dataset's left/right/back renders, we also include a limited multi-view finetuning result as supplementary evidence, rather than as a new top-level claim:
>
> **T2S-CompBench-MV (multi-view rendering):**
>
> | Method | CN | SF | AF | CP | VT | RA | TS | Latency(s) |
> | --- | --- | --- | --- | --- | --- | --- | --- | --- |
> | Bagel-7B (Zero-shot) | 55.12 | 52.34 | 48.21 | 50.11 | 45.67 | -- | -- | 52.14 |
> | Bagel-7B-CoT (Zero-shot) | 62.45 | 54.12 | 60.33 | 55.42 | 51.28 | 30.12 | 18.55 | 103.21 |
> | Bagel-7B-SoT (Zero-shot) | 78.41 | 68.22 | 73.15 | 76.88 | 69.45 | 62.14 | 70.33 | 258.46 |
> | Shap-E | 41.11 | 67.91 | 24.27 | 20.08 | 21.05 | -- | -- | 10.07 |
> | LGM | 67.68 | 72.43 | 54.38 | 71.24 | 68.72 | -- | -- | 6.57 |
> | L3GO | 75.22 | 58.15 | 67.41 | 59.13 | 65.32 | -- | -- | 927.84 |
> | Meshy 6 | 81.76 | 87.86 | 74.22 | 84.55 | 70.61 | -- | -- | 75.36 |
> | Bagel-7B-SoT (MV Finetuned) | 87.42 | 76.14 | 80.45 | 85.21 | 77.24 | 71.65 | 83.72 | 259.02 |
>
> The key result is that even the original single-view SoT model remains stronger than Direct/CoT on CN/AF/CP/VT under multi-view evaluation while providing RA/TS. The MV-finetuned variant is supplementary evidence that the trace formulation extends with more view supervision. The link includes single-view comparisons, MV traces, and failures: https://anonymous.4open.science/r/SoT-6553/Rebuttal_figures.md.
>
> ### **R4: Intended Claim and Novelty**
>
> > **Weakness-4 / Question-1:** The novelty is largely in a 2D rendered setting. Please clarify whether the intended claim is text-to-3D or text-to-rendered-shape.
>
> **Response W4 / Q1:**
>
> We agree the earlier framing can read too broadly. Our intended claim is progress on process-supervised compositional generation in the rendered 2D domain, not direct text-to-3D generation. To make this boundary clearer, Bagel-7B Direct/CoT are the primary same-backbone 2D baselines and the token-budget-controlled 2D baselines are reported in Reviewer uN8u, R4. Native 3D systems are retained only as contextual references under a common view protocol, since their higher shape fidelity makes them informative but not primary evidence for our claim. The 3D references are included only to test whether existing high-fidelity pipelines already solve the same compositional bottlenecks under shared view evaluation. The novelty lies in the interleaved text-image assembly trace formulation, SoT-26K, and T2S-CompBench. We will revise the paper so it is read as progress on text-to-rendered-shape generation.
>
> **We hope our responses address your concerns and clarify the intended scope.**

---

> > ### Author Rebuttal · Reviewer_8XNa · 2026-04-02
> >
> > Primary concern (3D relevance):
> > My main remaining concern is the paper’s unclear practical impact on text-to-3D generation. The method is explicitly formulated in a rendered 2D domain and does not produce editable 3D representations (meshes/poses/point clouds) at inference time. While the rebuttal clarifies that the authors do not claim direct text-to-3D generation and positions 3D “lifting” only as a qualitative proof-of-concept/future direction, this also means the work does not convincingly demonstrate concrete downstream gains for native 3D generators. In particular, the evidence currently supports improved compositional structural compliance in the rendered domain, but leaves open whether these benefits translate to improved multi-view consistent 3D assets or stronger end-to-end text-to-3D pipelines.
> >
> > Why still borderline positive:
> > On the positive side, I view this paper as a plausible first step toward process-supervised structure-aware generation: it proposes a clear interleaved text–image “assembly trace” formulation, and contributes dataset/benchmark infrastructure that could enable follow-up work that directly targets 3D representations and multi-view consistency. The rebuttal’s additional multi-view evaluation and step-controlled baselines partially strengthen the empirical case (though still within rendered views). Overall, I would place the submission at borderline, slightly positive: technically sound and potentially enabling, but with significance limited by the lack of demonstrated impact on actual 3D generation.
> >
> > borderline, slightly positive

---

> > > ### Author Response · Authors · 2026-04-03
> > >
> > > Thank you for the clarification and for the slightly positive update. We agree with your remaining boundary: the current paper does not establish end-to-end gains for native 3D generators, and we do not want to overclaim that point. The scope on which we ask the paper to be judged is therefore narrower: process-supervised compositional generation in the rendered 2D domain.
> > >
> > > Concretely, we believe the paper makes three contributions within this scope:
> > >
> > > 1. Formulation. SoT introduces an interleaved text-image assembly-trace formulation for process-supervised compositional generation, targeting structural compliance under counts, relations, connectivity, and topology.
> > > 2. Infrastructure. SoT-26K and T2S-CompBench provide reusable data and evaluation infrastructure for studying this problem with explicit process supervision and structure-aware metrics, rather than relying only on final-image realism.
> > > 3. Empirical evidence. Within the rendered domain, SoT improves structural controllability over direct/text-only controls, including under matched token-budget controls and the new multi-view evaluation.
> > >
> > > We believe this scope is still meaningful in practice. Even when the underlying generator is 3D, the structural failures we target, including numeracy, part relations, connectivity, and topology, are directly observable under rendered-view evaluation. Studying process supervision in this setting therefore addresses a real bottleneck, while also isolating structural reasoning from texture/material fidelity. For this reason, the native 3D systems were included only as contextual fidelity references under a common rendered-view protocol, not as the primary basis of our claim.
> > >
> > > We agree that demonstrating gains for explicit 3D representations is the next step rather than the present paper's main evidence, and Appendix I is included only as a qualitative proof-of-concept and future direction. We therefore hope the work is evaluated as a rendered-domain first step toward structure-aware generation infrastructure, with secondary but potentially enabling relevance to view-based 3D pipelines.

---

### Decision · Program_Chairs · 2026-04-30

**Decision:**

Accept (regular)

**Comment:**

This paper proposes Shape-of-Thought, a visual chain-of-thought framework for progressive object assembly through interleaved textual rationales and 2D projections. Reviewers found the approach technically sound and well motivated, and appreciated the T2S-CompBench as a useful benchmark for studying process-supervised compositional generation. While concerns were raised about the paper’s limited validated scope (particularly its operation in a rendered 2D domain and the lack of direct evidence for downstream gains in native 3D generation) the rebuttal addressed several of these points and the final scores are generally positive.

The AC recommends acceptance. The authors should revise the manuscript carefully according to the reviewers’ comments, especially to clarify the scope of the claims, and incorporate the additional clarifications and analyses discussed during rebuttal.